# The MADS-box transcription factor PHERES1 controls imprinting in the endosperm by binding to domesticated transposons

Rita A Batista[1,2], Jordi Moreno-Romero[1,2†], Yichun Qiu[1,2], Joram van Boven[1,2], Juan Santos-González[1,2], Duarte D Figueiredo[1,2‡], Claudia Köhler[1,2]*

[1]Department of Plant Biology, Uppsala BioCenter, Swedish University of Agricultural Sciences, Uppsala, Sweden; [2]Linnean Centre for Plant Biology, Swedish University of Agricultural Sciences, Uppsala, Sweden

**\*For correspondence:**
claudia.kohler@slu.se

**Present address:** [†]Centre for Research in Agricultural Genomics (CRAG), CSIC-IRTA-UAB-UB, Campus UAB, Barcelona, Spain; [‡]Institute for Biochemistry and Biology, University of Potsdam, Potsdam, Germany

**Competing interests:** The authors declare that no competing interests exist.

**Abstract** MADS-box transcription factors (TFs) are ubiquitous in eukaryotic organisms and play major roles during plant development. Nevertheless, their function in seed development remains largely unknown. Here, we show that the imprinted *Arabidopsis thaliana* MADS-box TF PHERES1 (PHE1) is a master regulator of paternally expressed imprinted genes, as well as of non-imprinted key regulators of endosperm development. PHE1 binding sites show distinct epigenetic modifications on maternal and paternal alleles, correlating with parental-specific transcriptional activity. Importantly, we show that the CArG-box-like DNA-binding motifs that are bound by PHE1 have been distributed by RC/Helitron transposable elements. Our data provide an example of the molecular domestication of these elements which, by distributing PHE1 binding sites throughout the genome, have facilitated the recruitment of crucial endosperm regulators into a single transcriptional network.

## Introduction

MADS-box transcription factors (TFs) are present in most eukaryotes, and are classified into two groups: type I or SRF (Serum Response Factor)-like, and type II or MEF2 (Myocyte Enhancing Factor2)-like (*Gramzow and Theissen, 2010*). In flowering plants, type I MADS-box TFs are associated with reproductive development and many are active in the endosperm, a nutritive seed tissue that supports embryo growth (*Bemer et al., 2010*). The endosperm is derived from the union of the homodiploid central cell with a haploid sperm cell. Therefore, this structure is a triploid tissue, composed of two maternal (M) and one paternal (P) genome copies. This 2M:1P genome ratio is crucial for the correct development of the endosperm, and any changes to this balance, for example upon hybridization of plants with different ploidies, lead to dramatic seed abortion phenotypes in a wide range of plant species (*Brink and Cooper, 1947*; *Håkansson, 1953*; *Lin, 1984*; *Scott et al., 1998*; *Sekine et al., 2013*). These phenotypes are endosperm-based, and stem from a deregulation in the process of endosperm cellularization – a crucial developmental transition in seed development (*Lafon-Placette and Köhler, 2014*). Hence, this endosperm-based seed abortion in response to interploidy hybridizations effectively establishes a reproductive barrier between newly polyploidized individuals and their ancestors, thus contributing to plant speciation (*Köhler et al., 2010*).

This aberrant endosperm development phenotype is observed both in interploidy and interspecies hybridizations, and has been linked to the deregulation of type I MADS-box TF genes both in the Brassicaceae (*Erilova et al., 2009*; *Lu et al., 2012*; *Rebernig et al., 2015*; *Tiwari et al., 2010*; *Walia et al., 2009*) and in crop species such as tomato and rice (*Ishikawa et al., 2011*; *Roth et al.,*

*2019*). In *Arabidopsis thaliana,* as well as in other species, the activity of type I MADS-box TFs is associated with the timing of endosperm cellularization: crosses in which the maternal parent has higher ploidy (maternal excess cross; e.g. 4x ♀ x 2x ♂) show early cellularization and downregulation of type I MADS-box TFs; whereas the opposite happens in paternal excess crosses (e.g. 2x ♀ x 4x ♂), where endosperm cellularization is delayed or non-occurring and type I MADS-box genes are dramatically upregulated (*Erilova et al., 2009*; *Kang et al., 2008*; *Lu et al., 2012*; *Tiwari et al., 2010*). Nevertheless, these observations have remained correlative, and a mechanistic explanation clarifying the role of MADS-box TFs in endosperm development remains to be established.

In this work, we characterized the function of the type I MADS-box TF PHERES1 (PHE1). *PHE1* is active in the endosperm and is a paternally expressed imprinted gene (PEG) (*Köhler et al., 2005*). Imprinting is defined as an epigenetic phenomenon that causes a gene to be expressed preferentially from the maternal or the paternal allele. It relies on parental-specific epigenetic modifications, which are asymmetrically established during male and female gametogenesis, and inherited in the endosperm (*Gehring, 2013*; *Rodrigues and Zilberman, 2015*). Demethylation of repeat sequences and transposable elements (TEs) – which occurs in the central cell, but not in sperm – is a major driver of imprinted gene expression (*Gehring, 2013*; *Rodrigues and Zilberman, 2015*). In maternally expressed genes (MEGs), DNA hypomethylation of maternal alleles leads to their expression, while DNA methylation represses the paternal allele (*Gehring, 2013*; *Rodrigues and Zilberman, 2015*). On the other hand, in PEGs, the hypomethylated maternal alleles undergo trimethylation of lysine 27 on histone H3 (H3K27me3), a repressive histone modification established by Fertilization Independent Seed-Polycomb Repressive Complex2 (FIS-PRC2). This renders the maternal alleles inactive, while the paternal allele is expressed (*Gehring, 2013*; *Moreno-Romero et al., 2016*; *Rodrigues and Zilberman, 2015*).

Similar to what has been observed for type I MADS-box TFs, imprinted genes have been implicated in correct endosperm development (*Erilova et al., 2009*; *Figueiredo et al., 2015*; *Jullien and Berger, 2010*; *Pignatta et al., 2018*; *Wolff et al., 2015*), and many PEGs have been shown to be upregulated strongly in the abortive endosperm of paternal excess cross seeds (*Wolff et al., 2015*). *PHE1*, being both a type I MADS-box TF gene and a PEG, constitutes an interesting case that can be used to explore further the relationship between the deregulation of these types of genes and the failure of endosperm development. A better understanding of this relationship could, in turn, contribute to solving the long-standing question of why these genes are essential for correct endosperm development.

Here, we identify PHE1 as a key transcriptional regulator of imprinted genes, of other type I MADS-box TFs, and of genes that are required for endosperm proliferation and cellularization. We show that deregulation of these genes in the endosperm of interploidy crosses is a direct consequence of *PHE1* upregulation. These results indicate that PHE1 is a central player in establishing a reproductive barrier between individuals of different ploidies, and provide mechanistic insight into how this is achieved. Furthermore, we explore the cross-talk between epigenetic regulation and PHE1 transcriptional activity at imprinted gene loci, and show that differential parental epigenetic modifications of PHE1 binding sites correlate with the transcriptional status of the parental alleles. Finally, we reveal that RC/Helitron TEs have served as distributors of PHE1 DNA-binding sites, providing an example of the molecular domestication of TEs.

## Results

### Identification of PHE1 target genes

To identify the genes that are regulated by PHE1, we performed a ChIP-seq experiment using siliques from a *PHE1::PHE1–GFP* line, which has been previously shown to express PHE1–GFP exclusively in the endosperm (*Weinhofer et al., 2010*). ChIP experiments were done using two biological replicates, and peaks were called independently in each replicate, using MACS2 (*Zhang et al., 2008*) (*Table 1*). Only peak regions that were common between the two replicates were considered for further analysis. These regions correspond to 2494 ChIP-seq peaks, which are henceforth referred to as PHE1 binding sites (*Table 2*). Annotation of these sites for genomic features revealed that the majority are located in promoter regions (*Table 2*), and that the highest density is detected 200–250 bp upstream of the transcriptional start site (TSS) (*Figure 1a*), similarly to what has been described

**Table 1.** PHE1 ChIP-seq read mapping and peak calling information.

Peak calling was done using the ChIP sample and its respective Input sample as control. The fraction of peaks present in both replicates was determined as the percentage of peaks for which spatial overlap between Replicate 1 and Replicate 2 peaks is observed (see Materials and methods).

| Sample | No. of sequenced reads | % of mapped reads | No. of called ChIP-seq peaks | % of ChIP-seq peaks present in both replicates |
|---|---|---|---|---|
| Replicate 1 *PHE1::PHE1–GFP* ChIP | 17,037,975 | 65.3 | | |
| Replicate 1 *PHE1::PHE1–GFP* Input | 24,276,095 | 71.1 | 2818 | 88.5 |
| Replicate 2 *PHE1::PHE1–GFP* ChIP | 21,838,147 | 70.5 | | |
| Replicate 2 *PHE1::PHE1–GFP* Input | 23,372,778 | 70.7 | 4521 | 55.2 |

for other *Arabidopsis* TFs (*Yu et al., 2016*). To avoid the identification of false-positives, only binding sites located up to 1.5 kb upstream and 0.5 kb downstream of the TSS were considered for the annotation of target genes. This corresponds to 83% of the PHE1 binding sites (*Figure 1a*), and allowed the identification of 1694 PHE1 target genes (*Table 2*, *Figure 1—figure supplement 1* and *Figure 1—source data 1*).

To further validate the target status of these genes, as well as to evaluate the impact of PHE1 in their transcriptional output, we assessed how their expression varies in response to changes in *PHE1* expression. Because *PHE1* is highly expressed in the endosperm of early seeds, and shows a decrease in expression as seed development progresses (*Figure 1b*), the influence of PHE1 activity on its targets can be assessed by monitoring their expression trend throughout seed growth. To achieve this, we sampled the endosperm expression level of all PHE1 targets from pre-globular to mature seed stages, using previously published transcriptome data for laser-capture microdissected endosperm (*Belmonte et al., 2013*) (*Figure 1c*). Using a *k*-means clustering analysis, we identified three different clusters of PHE1 targets, which showed distinct expression trends during seed development (*Figure 1c*, *Figure 1—source data 1*): genes in cluster 1 showed reduced expression as seed development progresses, mimicking the expression pattern observed for *PHE1* (*Figure 1b*); genes in clusters 2 and 3 showed small expression changes, with cluster 2 consisting of mildly upregulated genes, and cluster 3 consisting of mildly downregulated genes (*Figure 1c*). The general trend in which the expression of PHE1 targets follows PHE1 expression (cluster 1), or continues during the later stages of seed development (when PHE1 expression ceases) (clusters 2 and 3), suggests that PHE1 probably acts as transcriptional activator.

It is well established that *PHE1* and other type I MADS-box TF genes are highly overexpressed in the endosperm of paternal excess seeds (*Erilova et al., 2009*; *Kradolfer et al., 2013*; *Lu et al., 2012*; *Schatlowski and Köhler, 2012*; *Tiwari et al., 2010*). In order to generate *PHE1-*

**Table 2.** Annotation of PHE1 ChIP-seq peaks within genomic features of interest.

Annotation for each individual replicate, as well as for common peaks, is presented. For target gene analysis, only common peaks located 1.5 kb upstream to 0.5 kb downstream of the TSS were considered (3[rd] row).

| Sample | Total no. of peaks | No. of peaks in −1.5 kb to +0.5 kb window around TSS | Average distance to nearest TSS (bp) | Associated genomic feature (% of peaks) | | | No. of targeted genes |
|---|---|---|---|---|---|---|---|
| | | | | Promoter | Gene body | Intergenic | |
| Replicate 1 peaks | 2818 | 2182 | 445 | 88.6 | 4.7 | 6.6 | 1985 |
| Replicate 2 peaks | 4521 | 3508 | 600 | 86.6 | 3.5 | 9.9 | 2971 |
| Common peaks (PHE1 binding sites) | 2494 | 1995 | 430 | 89.6 | 4.6 | 5.8 | 1694 |

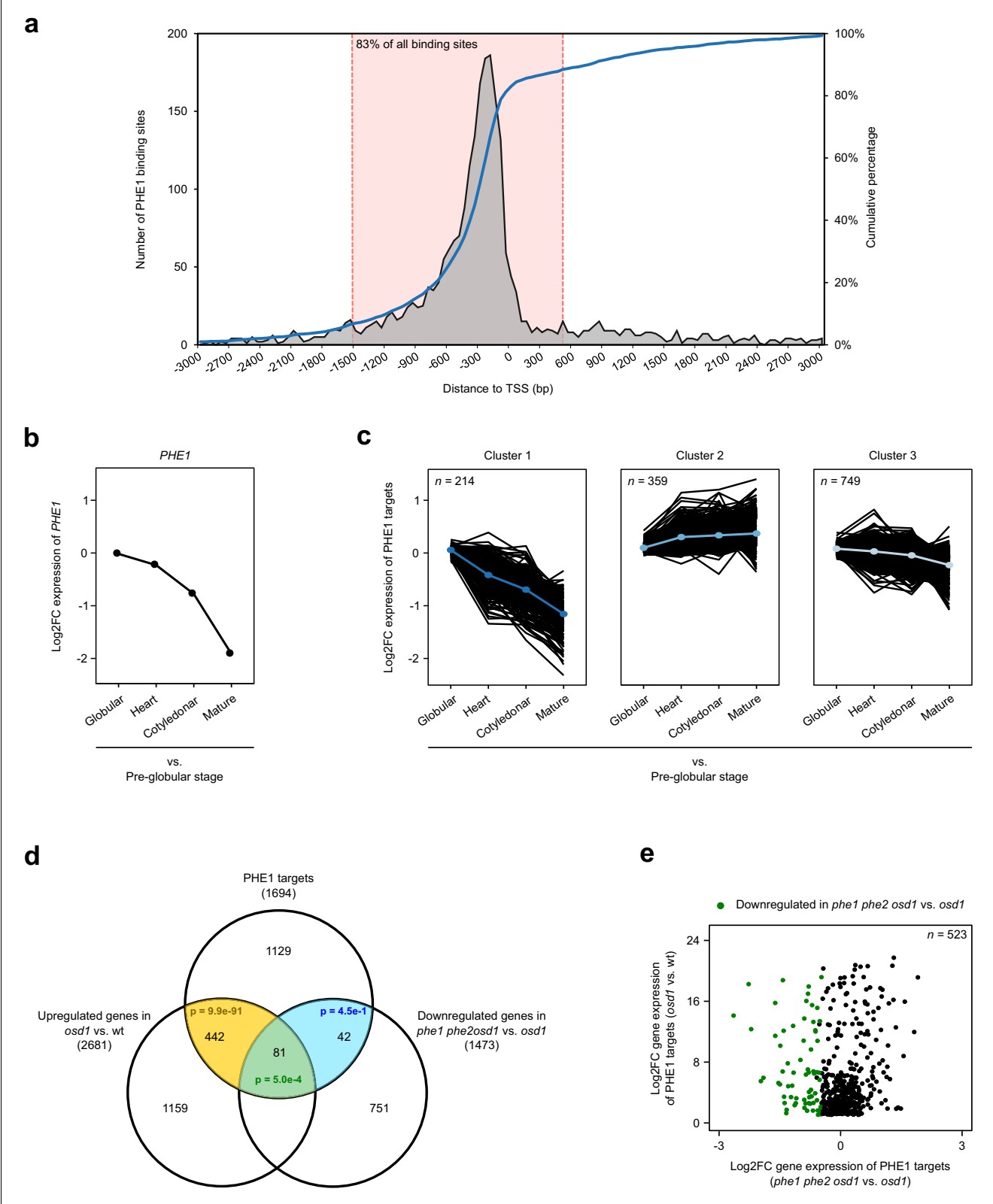

**Figure 1.** Identification and expression profile of PHE1 target genes. (**a**) Spatial distribution of PHE1 binding sites around transcription start sites (TSS). The dotted pink lines indicate the spatial interval used to define PHE1 target genes. (**b–c**) Expression of *PHE1* (**b**), and its target genes (**c**), across different stages of seed development. Gene expression is represented as a Log2-fold change between expression in the endosperm at the stages indicated on the x axis vs. expression in the pre-globular stage. A k-means clustering analysis was performed to group PHE1 targets that show similar

*Figure 1 continued on next page*

*Figure 1 continued*

expression trends across seed development. Gene expression data were retrieved from *Belmonte et al. (2013)*. (d) Overlap between PHE1 targets, genes that were found to be significantly upregulated in *osd1* when compared to wild-type (wt) seeds, and genes that were found to be significantly downregulated in *phe1 phe2 osd1* when compared to *osd1* seeds. P-values were determined using hypergeometric tests. (e) Expression of PHE1 targets that are significantly upregulated in *osd1* seeds when compared to wt seeds. Genes marked in green are also significantly downregulated in *phe1 phe2 osd1* seeds when compared to *osd1* seeds.

The online version of this article includes the following source data and figure supplement(s) for figure 1:

**Source data 1.** PHE1 target genes and their respective endosperm expression cluster.
**Figure supplement 1.** Examples of ChIP-seq read, peak, and motif distributions in PHE1 targets.
**Figure supplement 2.** Schematic representation of the *phe1 phe2* mutant.
**Figure supplement 3.** Expression of *PHE1* paralogs in *osd1* vs. wt and *phe1 phe2 osd1* vs. wt.

overexpressing seeds, we used the *omission of second division 1* (*osd1*) mutant, which produces diploid gametes at high frequency (*d'Erfurth et al., 2009*). We crossed *osd1* as pollen donor to a wild-type (wt) maternal plant (wt x *osd1*, abbreviated hereafter as *osd1*). This mimics a 2x♀ x 4x♂ cross, and results in a seed that is commonly denominated as triploid (3x), alluding to the 3x nature of the embryo. In parallel, we generated a *phe1* loss-of-function CRISPR/Cas9 mutant in the *phe2* background, as the two *PHE* genes are probably redundant (*Villar et al., 2009*) (*Figure 1—figure supplement 2*). We then introduced *phe1 phe2* into the *osd1* mutant background, and used this triple mutant as a pollen donor that was crossed to a wt maternal plant, thereby obtaining 3x seeds that lack PHE1 expression (wt x *phe1 phe2 osd1,* hereafter abbreviated as *phe1 phe2 osd1*). Thus, by generating transcriptomes of these seeds, and the control (wt x wt, hereafter abbreviated as wt), we could simultaneously assess the expression of PHE1 targets in response to the upregulation of *PHE1* expression (in *osd1* seeds) or the absence of *PHE1* expression (in *phe1 phe2 osd1* seeds) (*Figure 1d*).

As expected, we observed that a significant proportion of PHE1 targets were upregulated in *osd1* 3x seeds, correlating with increased expression of *PHE1*. Conversely, a moderate, but significant, number of PHE1 targets were downregulated in *phe1 phe2 osd1* when compared to *osd1* seeds (*Figure 1d*). Among these genes, 81 (p=5.0e–4) were simultaneously upregulated in *osd1* seeds and downregulated in *phe1 phe2 osd1* seeds (*Figure 1d–e*). We hypothesized that the small number of PHE1 target genes that show expression changes in *phe1 phe2 osd1* is probably a result of the redundant and compensatory activity of other type I MADS-box TFs in the endosperm of these seeds. Indeed, *PHE1* paralogs show a dramatic upregulation in *osd1* and *phe1 phe2 osd1* seeds, supporting this hypothesis (*Figure 1—figure supplement 3*). Because of this compensatory mechanism, we did not use this transcriptomics dataset to refine the list of PHE1 target genes obtained through ChIP-seq, as doing so would probably prevent the identification of biologically relevant genes. Nevertheless, the observation that a considerable proportion of genes show an endosperm expression pattern that mimics *PHE1* expression (*Figure 1b–c*), and also show upregulation upon PHE1 overexpression (*Figure 1d–e*), suggests that PHE1 acts as a transcriptional activator, and validates the binding sites obtained through ChIP-seq.

## PHE1 targets known regulators of endosperm development and other type I MADS-box transcription factors

To explore the functional role of PHE1 targets, we performed a Gene Ontology (GO) analysis (*Figure 2a*). This revealed several different enriched GO-terms associated with brassinosteroid signaling pathways, seed growth and development, and metabolic pathways including triglyceride biosynthesis and carbohydrate transport. Furthermore, we found an enrichment of a GO-term associated with the positive regulation of transcription. Indeed, many PHE1 targets are themselves transcriptional regulators (*Figure 2b*). More specifically, we detected that other type I MADS-box family genes are strongly over-represented among PHE1 targets (*Figure 2b*), pointing to a high degree of cross-regulation among members of this family. In addition, several PHE1 targets , such as *AGL62, YUC10, IKU2, MINI3,* and *ZHOUPI*, are known regulators of seed development (*Table 3*, *Figure 1—figure supplement 1*). These genes have been previously described to influence the proliferation, cellularization, and breakdown of the endosperm (*Table 3*), revealing an important role for PHE1 in regulating the development of this structure.

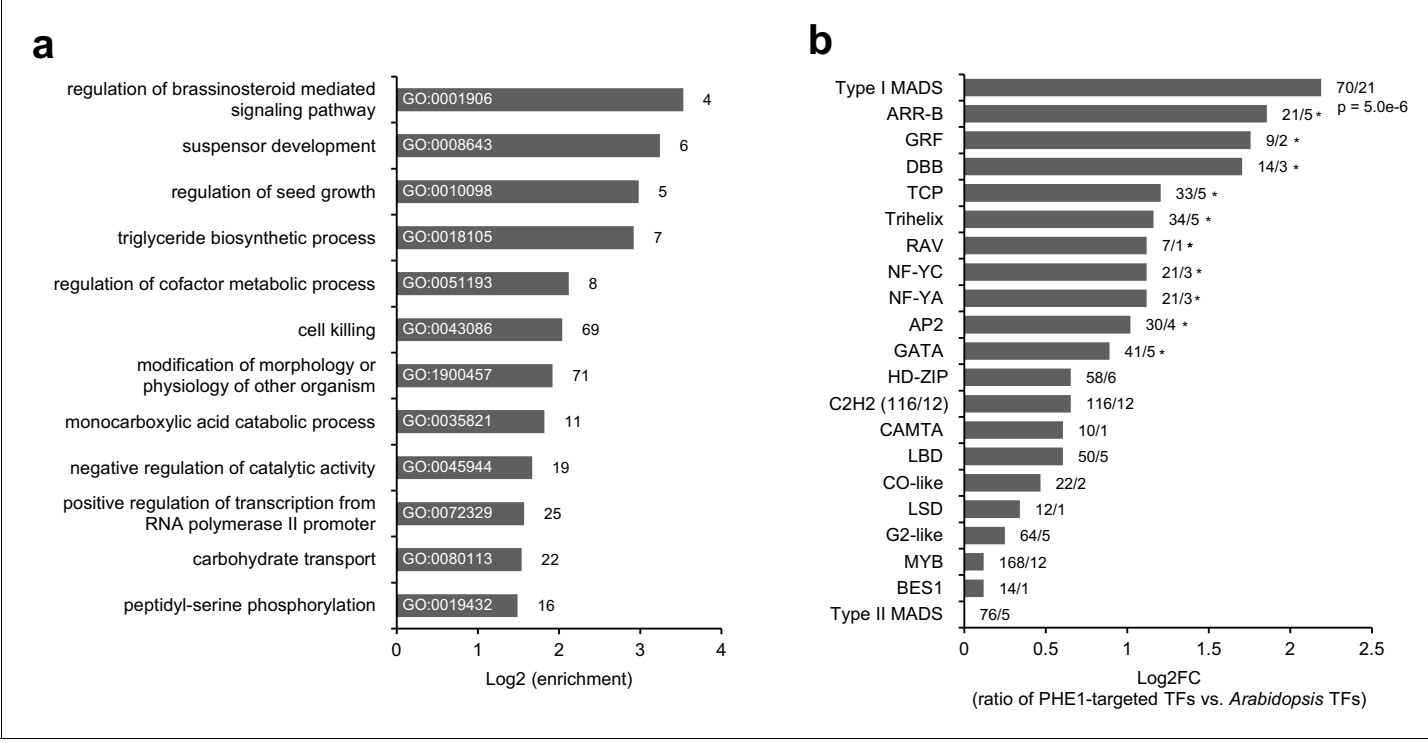

**Figure 2.** Transcription factor genes are enriched among PHE1 targets. (**a**) Enriched biological processes associated with PHE1 target genes. Numbers on bars indicate number of PHE1 target genes within each GO term. (**b**) Enrichment of transcription factor (TF) families among PHE1 target genes (see Materials and methods). Numbers indicate the total number of *Arabidopsis* genes belonging to a certain TF family, and the total number of genes in that family targeted by PHE1. *, p-values <0.05. P-values were determined using the hypergeometric test.

## Transposable elements act as cis-regulatory elements by carrying PHE1 DNA-binding motifs

To investigate the DNA-binding properties of PHE1, we screened PHE1 binding sites for enriched sequence motifs, using HOMER's de novo motif analysis tool. This analysis revealed that PHE1 uses two distinct DNA-binding motifs: motif A was present in about 53% of PHE1 binding sites, and motif B could also be found in 43% of those sites (*Figure 3a*). In total, 68% of all of PHE1 binding sites are associated with at least one of these motifs, and the majority of these show co-occurrence of Motif A and B (*Figure 3—figure supplement 1a*). In addition, and as observed for type II MADS-box TFs (*Aerts et al., 2018*), the highest density of these motifs is detected at the center of PHE1 binding sites (*Figure 3—figure supplement 1b*). PHE1 DNA-binding motifs closely resemble CArG-boxes, the signature motif of binding sites for type II MADS-box TFs (*de Folter and Angenent, 2006*) (*Figure 3a*, *Figure 3—figure supplement 1c*). This is particularly visible in the case of Motif B, which is similar to the SEP3 DNA-binding motif, and shows the characteristics of a *bona fide* CArG-box -– CC(A/T)$_6$GG (*Figure 3—figure supplement 1c*). Motif A, on the other hand, shows similarity to the SVP CArG-box, but lacks the terminal G nucleotides, thus resembling a partially degenerated CArG-box (*Figure 3—figure supplement 1c*). Nevertheless, the detected similarity between PHE1 DNA-binding motifs and previously characterized CArG-boxes suggests that the DNA-binding properties of type I and type II TFs might be conserved.

Strikingly, we observed that PHE1 binding sites significantly overlap with TEs, preferentially with those of the RC/Helitron superfamily (29% in PHE1 binding sites, versus 9% in random binding sites) (*Figure 3b*), and in particular with some RC/Helitron subfamilies (*Figure 3—figure supplement 2*). We thus addressed the question of whether RC/Helitrons contain sequence properties that promote PHE1 binding. Indeed, a screen of genomic regions for the presence of PHE1 DNA-binding motifs, revealed significantly higher motif densities within all RC/Helitrons, when compared to other TE superfamilies (*Figure 3c*). We found that most RC/Helitron sequences associated with PHE1 DNA-binding motifs share sequence homology, and that the majority of them could be grouped into one

**Table 3.** PHE1 target genes previously implicated in endosperm development.

| Gene ID | Imprinting status | Description of function |
|---|---|---|
| AGL62 | Non-imprinted | Type I MADS-box TF involved in endosperm proliferation and seed coat development (*Figueiredo et al., 2016*; *Figueiredo et al., 2015*; *Kang et al., 2008*; *Roszak and Köhler, 2011*) |
| YUC10 | PEG | Flavin monooxygenase that catalyzes the last step of the Trp-dependent auxin biosynthetic pathway (*Zhao, 2012*). Involved in endosperm proliferation and cellularization (*Batista et al., 2019*; *Figueiredo et al., 2015*) |
| IKU2 | Non-imprinted | Encodes a leucine-rich repeat receptor kinase protein that, together with *MINI3*, is part of the IKU pathway controlling seed size. *iku2* mutants show reduced endosperm growth and early endosperm cellularization (*Garcia et al., 2003*; *Luo et al., 2005*). |
| MINI3 | Non-imprinted | WRKY TF that, together with *IKU2*, is part of the IKU pathway controlling seed size. *mini3* mutants show reduced endosperm growth and early endosperm cellularization (*Luo et al., 2005*). |
| ZHOUPI | Non-imprinted | Encodes a bHLH TF expressed in the embryo-surrounding region of the endosperm. It is essential for embryo cuticle formation and endosperm breakdown after its cellularization (*Xing et al., 2013*; *Yang et al., 2008*). |
| MEA | MEG | Subunit of the FIS–PRC2 complex, responsible for depositing H3K27me3 at target loci including PEGs (*Moreno-Romero et al., 2016*; *Moreno-Romero et al., 2019*). Loss of MEA and, consequently, paternally biased expression of PEGs lead to a 3x-seed-like phenotype (*Grossniklaus et al., 1998*; *Kiyosue et al., 1999*). |
| ADM | PEG | Interacts with SUVH9 and AHL10 to promote H3K9me2 deposition in TEs, influencing the expression of neighboring genes. Mutations in *ADM* lead to rescue of the 3x seed abortion phenotype (*Jiang et al., 2017*; *Kradolfer et al., 2013*; *Wolff et al., 2015*). |
| SUVH7 | PEG | Encodes a putative histone-lysine N-methyltransferase. Mutations in *SUVH7* lead to rescue of the 3x seed abortion phenotype (*Wolff et al., 2015*). |
| PEG2 | PEG | Encodes an unknown protein, which is not translated in the endosperm. *PEG2* transcripts act as a sponge for siRNA854, thus regulating UBP1 abundance (*Wang et al., 2018*). Mutations in *PEG2* lead to rescue of the 3x seed abortion phenotype (*Wang et al., 2018*; *Wolff et al., 2015*). |
| NRPD1a | PEG | Encodes the largest subunit of RNA POLYMERASE IV, which is involved in the RNA-directed DNA methylation pathway. Mutations in *NRPD1a* lead to rescue of the 3x seed abortion phenotype (*Erdmann et al., 2017*; *Martinez et al., 2018*). |

large cluster on the basis of sequence identity (*Figure 3—figure supplement 3*). Although the presence of additional smaller clusters points to a few instances of independent gains of PHE1 DNA-binding motifs through de novo mutation or sequence capture, the grouping of most sequences within one cluster suggests that a single ancestral RC/Helitron probably acquired a perfect or nearly perfect motif. Moreover, we could detect the presence of perfect or nearly perfect PHE1 DNA-binding motifs within the consensus sequences of several RC/Helitron families (*Figure 3—figure supplement 4*), suggesting that the radiation of RC/Helitrons occurred after the acquisition of the binding motif.

Interestingly, even though motif densities were higher in RC/Helitrons that were overlapped by PHE1 binding sites than in non-overlapped RC/Helitrons, this difference was not significant (*Figure 3c*). As the enrichment of PHE1 DNA-binding motifs is a specific feature of RC/Helitrons, the domestication of these TEs as cis-regulatory regions might facilitate TF binding and the modulation of gene expression. In line with this, we detected that genes that are flanked by RC/Helitrons carrying bound PHE1 DNA-binding motifs were expressed to a greater level in the endosperm than in other seed tissues, and were expressed at levels similar to those of PHE1 targets without flanking RC/Helitrons (*Figure 3d*). This shows that for a subset of genes, TEs can be effectively used as sites for PHE1 binding, thus triggering the endosperm-specific expression of nearby genes.

## Epigenetic status of imprinted gene promoters conditions PHE1 accessibility in a parent-of-origin- specific manner

We detected a significant enrichment of imprinted genes among the PHE1 target genes, with 9% of all maternally expressed genes (MEGs) and 27% of all paternally expressed genes (PEGs) being targeted (*Figure 4a*, *Figure 4—source data 1*). Given the significant overrepresentation of imprinted genes among PHE1 targets, we assessed how the epigenetic landscape at those loci correlates with DNA-binding by PHE1. We surveyed levels of endosperm H3K27me3 within PHE1-binding sites and identified two distinct clusters (*Figure 4b*). Cluster 1 was characterized by an accumulation of H3K27me3 in regions flanking the center of the binding site, whereas the center itself was devoid of this mark. This pattern of distribution was maintained when taking into consideration the strand location of the gene associated with the binding site, revealing that binding of PHE1 at these sites

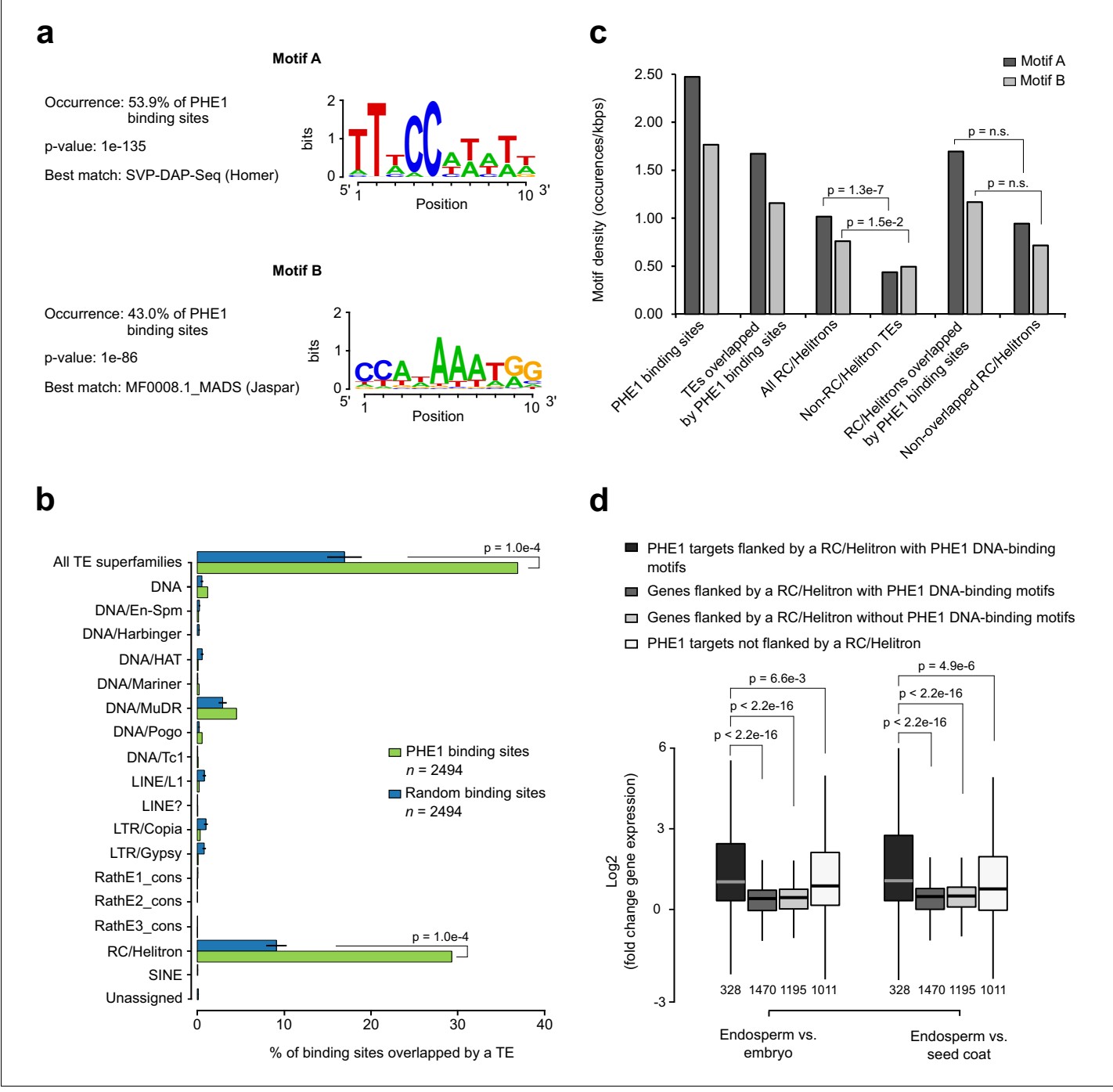

**Figure 3.** RC/Helitrons carry PHE1 DNA-binding motifs. (a) CArG-box like DNA-binding motifs identified from PHE1 ChIP-seq data. (b) Fraction of PHE1 binding sites (green) that overlap transposable elements (TEs). Overlap is expressed as the percentage of total binding sites for which spatial intersection with features on the y-axis is observed. A set of random binding sites is used as control (blue). This control set was obtained by randomly shuffling the identified PHE1 binding sites within random *A. thaliana* gene promoters (see Materials and methods). P-values were determined using Monte Carlo permutation tests (see Materials and methods). Bars represent ± s.d. (n = 2494, for PHE1 binding sites and random binding sites). (c) Density of PHE1 DNA-binding motifs in different genomic regions of interest. P-values were determined using $\chi^2$ tests. (d) Fold-change in the expression of genes flanked by RC/Helitron TEs. Fold-change was determined by comparing endosperm and embryo, or by comparing endosperm and seed coat. Genes were divided into four categories depending on their PHE1 target status, and the presence of RC/Helitrons with and without PHE1 DNA-binding motifs. Gene expression data were retrieved from *Belmonte et al. (2013)*. Pre-globular seed stage was used in this analysis. P-values were determined using two-tailed Mann-Whitney tests (n = number represented below boxplots).

The online version of this article includes the following figure supplement(s) for figure 3:

*Figure 3 continued on next page*

**Figure supplement 1.** PHE1 DNA-binding motifs show similarity to type II MADS-box CArG-boxes.
**Figure supplement 2.** RC/Helitron family overlap with PHE1 binding sites.
**Figure supplement 3.** Clustering of bound PHE1 DNA-binding motifs.
**Figure supplement 4.** PHE1 DNA-binding motif densities in RC/Helitron consensus sequences.

occurs in an H3K27me3-depleted island (*Figure 4—figure supplement 1*). Cluster 2, on the other hand, contained binding sites that are largely devoid of H3K27me3. The distribution of H3K27me3 in cluster 1 was mostly attributed to the deposition of H3K27me3 on the maternal alleles, whereas the paternal alleles were devoid of this mark (*Figure 4c*) — a pattern usually associated with PEGs (*Moreno-Romero et al., 2016*). Consistently, genes that are associated with cluster 1 binding sites had more paternally biased expression in the endosperm when compared to genes associated with cluster 2 (*Figure 4—figure supplement 2a*). This is reflected by the association of more PEGs and putative PEGs with cluster 1 (*Figure 4—figure supplement 2b*). We also identified parental-specific differences in DNA methylation, specifically in the CG context: PHE1-binding sites that were associated with MEGs had significantly higher methylation levels in paternal alleles than in maternal alleles (*Figure 4d*).

We hypothesized that the differential parental deposition of epigenetic marks in parental alleles of PHE1 binding sites can result in the differential accessibility of each allele. This might impact the binding of PHE1, and therefore also impact transcription in a parent-of-origin-specific manner. To test this, we performed ChIP using a L*er* maternal plant and a Col *PHE1::PHE1–GFP* pollen donor, taking advantage of single nucleotide polymorphisms (SNPs) between these two accessions to discern parental preferences of PHE1 binding. Using Sanger sequencing, we determined the parental origin of enriched ChIP-DNA in MEG, non-imprinted, and PEG targets (*Figure 4e*, *Figure 4—figure supplement 3*). Although binding of PHE1 was biallelic in non-imprinted targets (*Figure 4e*), only maternal binding was detected in the tested MEG targets (*Figure 4e*), supporting the idea that CG hypermethylation of paternal alleles prevents their binding by PHE1. Interestingly, we observed biallelic binding in PEG targets (*Figure 4e*). Even though the maternal PHE1 binding sites in PEGs were flanked by H3K27me3 (*Figure 4b–c*), correlating with the transcriptional repression of maternal alleles, the absence of this mark within the binding site centers seems to be permissive for maternal PHE1 binding.

## Insertion of transposable elements carrying PHE1 DNA-binding motifs correlates with gain of imprinting

Previous studies have shown that PEGs are often flanked by RC/Helitrons (*Hatorangan et al., 2016*; *Pignatta et al., 2014*; *Wolff et al., 2011*), a phenomenon that has been suggested to lead to the parental asymmetry of epigenetic marks in these genes (*Moreno-Romero et al., 2016*; *Pignatta et al., 2014*). Consistent with our finding that PHE1 binding sites overlapped with RC/Helitrons (*Figure 3b*), we found that PHE1 DNA-binding motifs were contained within these TEs significantly more frequently in PEGs than in non-imprinted genes (*Figure 5—figure supplement 1*). Furthermore, we detected the presence of homologous RC/Helitrons containing PHE1 binding motifs in the promoter regions of several PHE1-targeted PEG orthologs (*Figure 5a–e*, *Figure 5—figure supplement 2*), indicating ancestral insertion events. The presence of these RC/Helitrons correlated with paternally biased expression of the associated orthologs, providing further support to the hypothesis that these TEs contribute to the gain of imprinting, especially of PEGs.

## PHE1 establishes triploid seed inviability of paternal excess crosses

Among the PEGs that were targeted by PHE1 were *ADM*, *SUVH7*, *PEG2,* and *NRPD1a* (*Table 3*). Mutants in all four PEGs suppress the abortion of 3x seeds generated by paternal excess interploidy crosses (*Martinez et al., 2018*; *Satyaki and Gehring, 2019*; *Wolff et al., 2015*). Furthermore, we found that between 40% and 50% of highly upregulated genes in 3x seeds are targeted by PHE1 (*Figure 6a*), suggesting that this TF might play a central role in mediating the strong gene deregulation observed in these seeds. If this is true, removal of PHE1 in paternal excess 3x seeds is expected to suppress their inviability. Indeed, although wt and *phe2* maternal plants pollinated with *osd1*

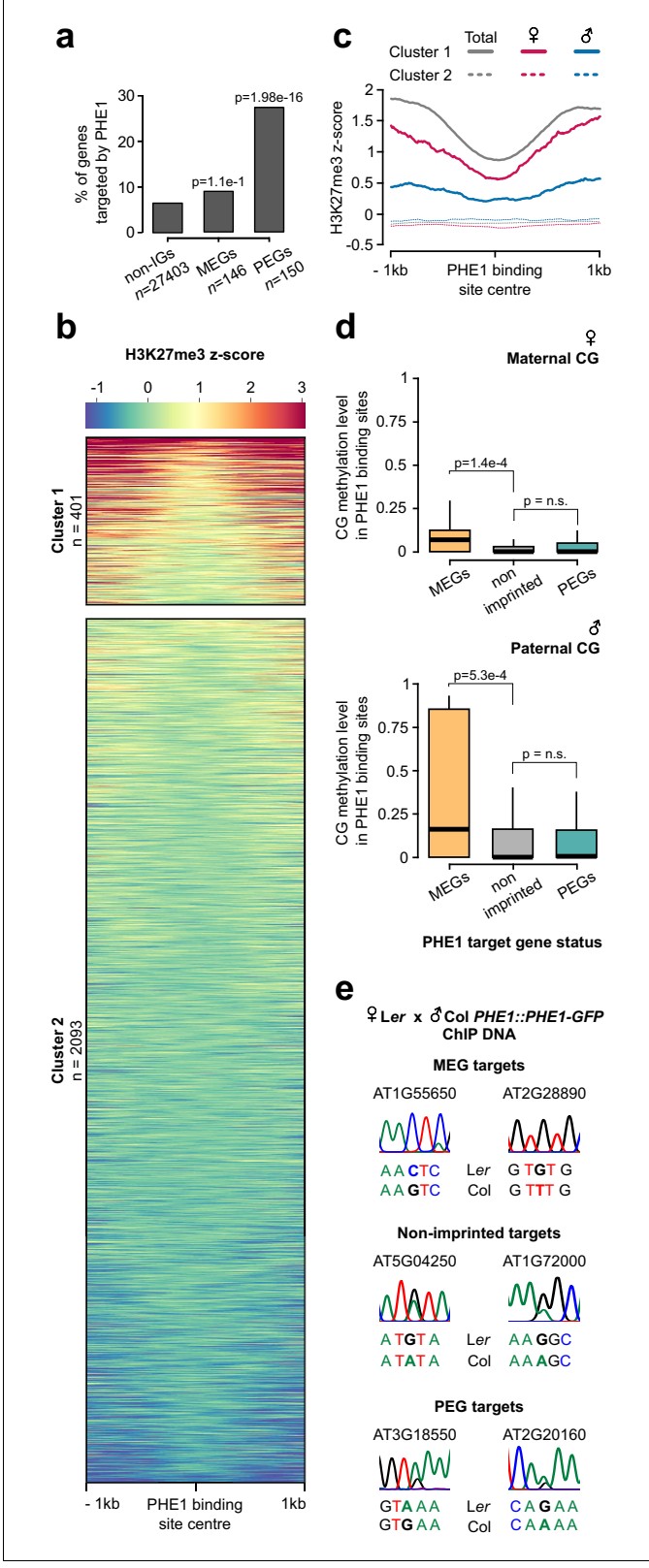

**Figure 4.** Parental asymmetry of epigenetic marks in imprinted gene promoters conditions PHE1 binding. (a) Fraction of non-imprinted and published imprinted genes targeted by PHE1. P-values were determined using the hypergeometric test. The list of published imprinted genes used for this analysis is detailed in *Figure 4—source data 1*. (b) Heatmap of endosperm H3K27me3 distribution along PHE1-binding sites. Each horizontal line

*Figure 4 continued on next page*

*Figure 4 continued*

represents one binding site. Clusters were defined on the basis of the pattern of H3K27me3 distribution (see Materials and methods) (c) Metagene plot of average maternal (♀, pink), paternal (♂, blue) and total (grey) endosperm H3K27me3 marks along PHE1 binding sites. (d) CG methylation levels in maternal (♀, upper panel) and paternal (♂, lower panel) alleles of PHE1 binding sites associated with MEGs (yellow), PEGs (green) and non-imprinted (grey) PHE1 targets. P-values were determined using two-tailed Mann-Whitney tests. (e) Sanger sequencing of imprinted and non-imprinted gene promoters bound by PHE1. SNPs for maternal (Ler) and paternal (Col) alleles are shown (*n* = 1 biological replicate). Maternal:total read ratios for imprinted genes are as follows: AT1G55650 – 1.0; AT2G28890 – 0.97; AT3G18550 – 0.44; and AT2G20160 – 0.32.

The online version of this article includes the following source data and figure supplement(s) for figure 4:

**Source data 1.** List of imprinted genes used in this study.
**Figure supplement 1.** Directionality of H3K27me3 distribution in Cluster 1 binding sites.
**Figure supplement 2.** Characterization of H3K27me3 clusters.
**Figure supplement 3.** Parental-specific PHE1 ChIP.

pollen form 3x seeds that abort at high frequency (*Kradolfer et al., 2013*), *phe1 phe2 osd1* pollen strongly suppresses 3x seed inviability (*Figure 6b*, *Figure 6—figure supplement 1a*). This is also reflected by the increased germination of 3x *phe1 phe2* seeds (*Figure 6c*, *Figure 6—figure supplement 1b*), and this phenotype could be reverted by introducing the *PHE1::PHE1–GFP* transgene paternally (*Figure 6b–c*, *Figure 6—figure supplement 1a–b*). Notably, 3x seed rescue was mostly mediated by *phe1*, as the presence of a wt *PHE2* allele in 3x seeds (wt x *phe1 phe2 osd1*) led to rescue levels that were comparable to those seen when no wt *PHE2* allele was present (*phe2* x *phe1 phe2 osd1*) (*Figure 6b–c*, *Figure 6—figure supplement 1a–b*). Importantly, *phe1*-mediated 3x seed rescue was accompanied by reestablishment of endosperm cellularization (*Figure 6—figure supplement 1c–d*).

## Imprinted gene deregulation in triploid seeds is not accompanied by a breakdown of imprinting

Loss of FIS-PRC2 function causes a phenotype similar to that of paternal excess 3x seeds, correlating with largely overlapping sets of deregulated genes, notably PEGs (*Erilova et al., 2009*; *Tiwari et al., 2010*). As FIS-PRC2 is a major regulator of PEGs in the *Arabidopsis* endosperm (*Moreno-Romero et al., 2016*), we addressed the question of whether imprinting is disrupted in 3x seeds. To assess this, we analyzed the parental expression ratio of imprinted genes in the endosperm of 2x and 3x seeds. Surprisingly, imprinting was not disrupted in 3x seeds (*Figure 6d*). To confirm that maintenance of imprinting in 3x seeds is explained by maintenance of the underlying epigenetic marks, we generated parental-specific H3K27me3 profiles of 2x and 3x seed endosperm (*Table 4*). Consistent with the observed maintenance of imprinting, we detected similar H3K27me3 levels on the maternal alleles of PEGs in 2x and 3x seeds (*Figure 6e*, *Figure 6—figure supplement 2*). Collectively, these data show that the major upregulation of PEG expression in 3x seeds is due to increased transcription of the active allele, probably mediated by PHE1 and other MADS-box TFs, with maintenance of the imprinting status.

## Discussion

### Functional role of PHE1 during endosperm development

*PHE1* is expressed during the early stages of seed development, immediately after fertilization, and until the onset of endosperm cellularization. The initial stages of endosperm development are characterised by a rapid proliferation that is accompanied by import of resources to this tissue (*Hill et al., 2003*; *Mansfield and Briarty, 1990*; *Morley-Smith et al., 2008*). Upon cellularization, these resources are hypothesized to be transferred to the embryo, correlating with rapid growth and the accumulation of storage products in this structure (*Baud et al., 2008*; *Hehenberger et al., 2012*; *Hill et al., 2003*). Remarkably, we found that genes that are required for endosperm proliferation and growth, such as *YUC10* (*Figueiredo et al., 2015*), *IKU2* (*Garcia et al., 2003*; *Luo et al., 2005*), and *MINI3* (*Luo et al., 2005*) are under transcriptional control of PHE1. As *PHE1* is imprinted

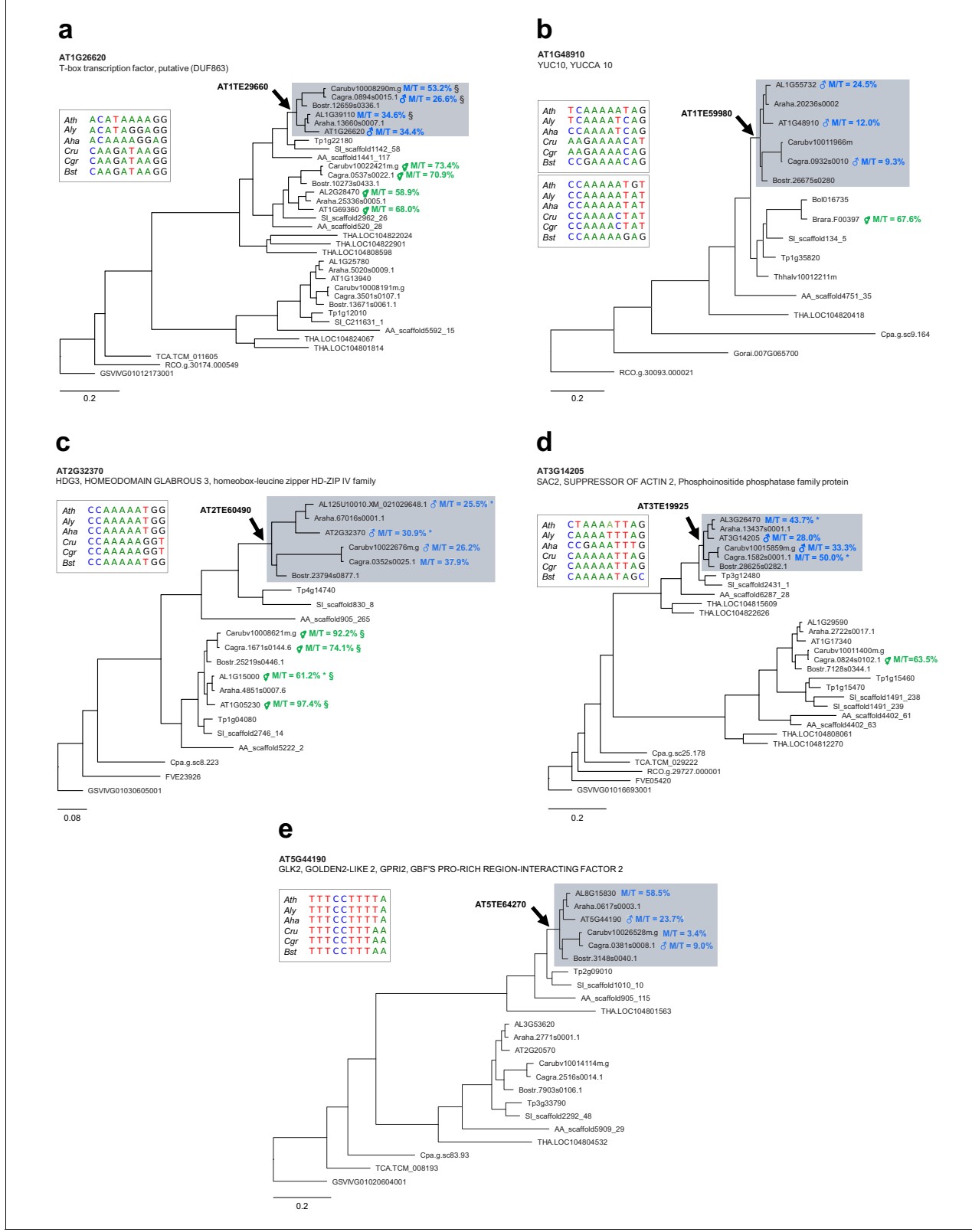

**Figure 5.** Ancestral RC/Helitron insertions are associated with gain of imprinting in the Brassicaceae. (**a–e**) Phylogenetic analyses of PHE1-targeted PEGs and their homologs. Each panel represents a distinct target gene and its corresponding homologs in different species. The genes shown on a grey background have homologous RC/Helitron sequences in their promoter region. The arrow indicates the putative insertion of an ancestral RC/Helitron. The identity of the RC/Helitron identified in *A. thaliana* is indicated. These *A. thaliana* RC/Helitrons contain a PHE1 DNA-binding motif and are

*Figure 5 continued on next page*

*Figure 5 continued*

associated with a PHE1 binding site. The inset boxes represent the alignment between the *A. thaliana* PHE1 DNA-binding motif and similar DNA motifs contained in RC/Helitrons that are present in the promoter regions of orthologous genes. When available, the imprinting status of a given gene is indicated by the presence of ♂ (PEG) or ♀ (non-imprinted), and reflects the original imprinting analyses done in the source publications (see Materials and methods). The maternal:total read ratio (M/T) for each gene is also indicated. §: potential contamination from maternal tissue. *: accession-biased expression. The scale bars represent the frequency of substitutions per site for the ML tree. The tree is unrooted. Gene identifier nomenclatures: AT, *Arabidopsis thaliana*; AL, *Arabidopsis lyrata*; Araha, *Arabidopsis halleri*; Bostr, *Boechera stricta*; Carubv, *Capsella rubella*; Cagra, *Capsella grandiflora*; Tp, *Schrenkiella parvula*; SI, *Sisymbrium irio*; Bol, *Brassica oleracea*; Brapa, *Brassica rapa*; Thhalv, *Eutrema salsugineum*; AA, *Aethionema arabicum*; THA, *Tarenaya hassleriana*; Cpa, *Carica papaya*; TCA, *Theobroma cacao*; Gorai, *Gossypium raimondii*; RCO, *Ricinus communis*; FVE, *Fragaria vesca*; and GSVIVG, *Vitis vinifera*.

The online version of this article includes the following figure supplement(s) for figure 5:

**Figure supplement 1.** RC/Helitrons carrying PHE1 DNA-binding motifs are more prevalent in PEGs.
**Figure supplement 2.** Analysis of PHE1-targeted PEG orthologs and upstream RC/Helitron sequences within Brassicaceae.

and paternally expressed (*Köhler et al., 2005*), proliferation of the endosperm becomes dependent on the presence of the paternal genome, which allows the fertilization event to be coupled with the onset of endosperm growth. Besides being relevant for endosperm proliferation, PHE1 might also have an important role in resource accumulation in preparation for cellularization, because genes that are related to metabolic processes, such as carbohydrate transport and triglyceride biosynthesis, were found to be enriched among PHE1 targets.

The control of both endosperm proliferation and resource accumulation by PHE1 suggests that type I MADS-box TFs might have a central role in establishing essential transcriptional networks that facilitate endosperm development. Besides PHE1, many other type I MADS-box TFs are expressed in the endosperm (*Bemer et al., 2010*), and probably act as heterodimers (*de Folter et al., 2005*). Similarly to the function of type II MADS-box TFs in the determination of flower morphology (*Chen et al., 2018*; *Smaczniak et al., 2017*; *Theissen and Saedler, 2001*), different type I MADS-box heterodimers could be able to control distinct sets of genes during endosperm development. Furthermore, different stages of endosperm development could be characterized by the activity of different heterodimers. Therefore, a deeper analysis of the expression and interaction dynamics of type I MADS-box TFs might reveal that this family has a role in endosperm development that is broader than that uncovered here for PHE1.

## Deregulated expression of type I MADS-box transcription factors establishes hybridization barriers

Normal endosperm development is associated with decreased expression of a subset of type I MADS-box genes, including *PHE*, preceding the onset of cellularization (*Erilova et al., 2009*; *Lu et al., 2012*; *Schatlowski and Köhler, 2012*; *Stoute et al., 2012*; *Tiwari et al., 2010*; *Walia et al., 2009*). This has led to the suggestion that these TFs are negative regulators of endosperm cellularization, although the molecular mechanism behind these observations has remained elusive. Our observation that PHE1 controls the expression of a large fraction of deregulated genes in interploidy hybridization seeds, in which cellularization is disturbed, provides an explanation for why the onset of endosperm cellularization is correlated with the expression of *PHE1*. Because *PHE1* expression is increased in the endosperm of paternal excess crosses, the expression of PHE targets is probably similarly affected. In the specific case of the target gene *YUC10*, this leads to prolonged auxin biosynthesis and to the overaccumulation of this hormone in the endosperm, which has been previously shown to prevent the onset of cellularization (*Batista et al., 2019*). Therefore, successful and adequately timed endosperm cellularization probably requires a specific pattern of expression of type I MADS-box genes, as well as a balanced stoichiometry between members of different MADS-box protein complexes.

Interploidy and interspecies hybridizations can easily change the expression and stoichiometry of MADS-box TFs, for example by having distinct numbers of gene copies, or through differences in gene regulation between parents, such as variations in imprinting patterns (*Dilkes and Comai, 2004*). Therefore, it is possible that type I MADS-box TFs can function as sensors of parental compatibility in the endosperm: i) these TFs show expression differences upon interploidy/interspecies hybridization (*Erilova et al., 2009*; *Lu et al., 2012*; *Schatlowski and Köhler, 2012*; *Stoute et al.,*

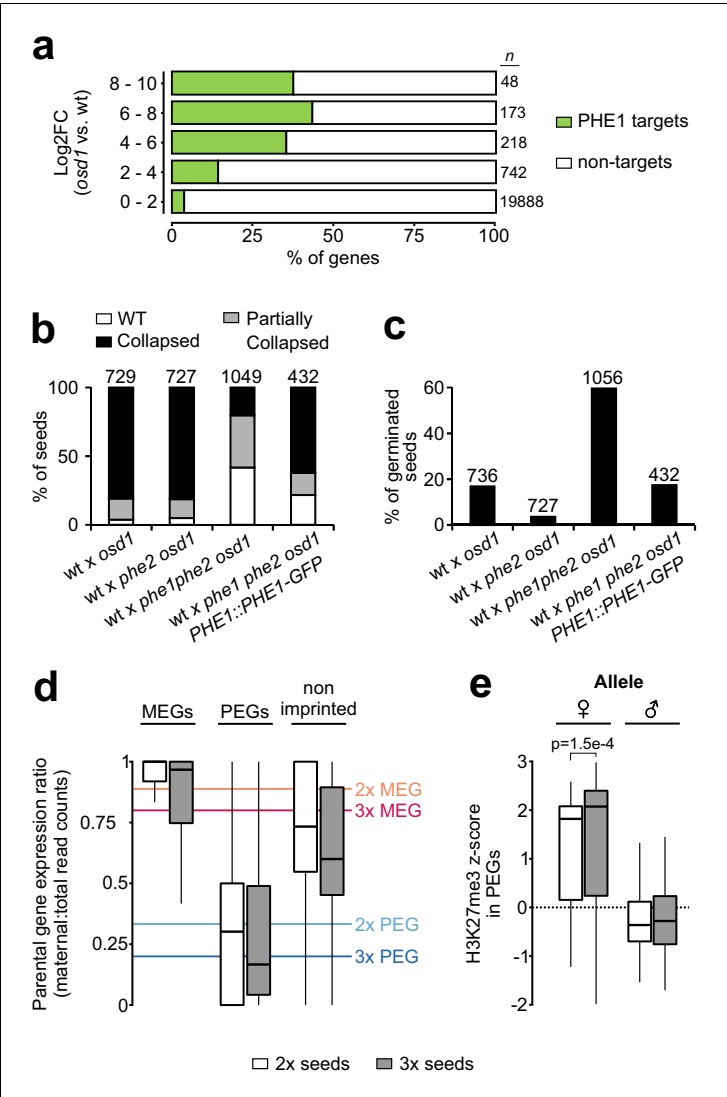

**Figure 6.** PHE1 establishes 3x seed inviability of paternal excess crosses. (a) Target status of upregulated genes in paternal excess crosses. Highly upregulated genes in 3x seeds are more often targeted by PHE1 (p<2.2e–16, $\chi^2$ test). $n$ = numbers on the left. (b, c) Seed inviability phenotype (b) of paternal excess crosses in wild-type (wt), *phe1 phe2*, and *phe1* complementation lines, with their respective seed germination rates (c). The maternal parent is always indicated first. Remaining control crosses are shown in *Figure 6—figure supplement 1a–b*. $n$ = numbers on top of bars (seeds). (d) Parental expression ratio of imprinted genes in the endosperm of 2x (white) and 3x seeds (grey). Solid lines indicate the ratio thresholds for the definition of MEGs and PEGs in 2x and 3x seeds. (e) Accumulation of H3K27me3 across maternal (♀) and paternal (♂) gene bodies of PEGs in the endosperm of 2x and 3x seeds (white and grey, respectively). H3K27me3 accumulation in MEGs and non-imprinted genes is shown in *Figure 6—figure supplement 2*. P-value was determined using a two-tailed Mann-Whitney test.

The online version of this article includes the following figure supplement(s) for figure 6:

**Figure supplement 1.** Rescue of 3x seed inviability in *phe1 phe2*.

**Figure supplement 2.** Distribution of H3K27me3 in 2x and 3x seeds.

2012; *Tiwari et al., 2010*; *Walia et al., 2009*); and ii) as exemplified here by PHE1, they are able to engage pathways that enforce seed abortion when compatibility between parents is not present. Dissecting the factors governing expression of type I MADS-box TFs will be an important future step in better understanding the regulatory mechanisms triggering endosperm-based reproductive barriers.

**Table 4.** H3K27me3 ChIP-seq read mapping and purity information.

| Sample | No. trimmed reads | % of mapped reads | No. of Ler reads | No. of Col reads | Purity (%) |
|---|---|---|---|---|---|
| Ler x Col 4x Replicate 1 Input | 30,337,933 | 68.3 | 1,844,412 | 1,563,394 | 95.7 |
| Ler x Col 4x Replicate 1 H3 ChIP | 22,644,505 | 73.3 | 1,439,255 | 1,211,446 | |
| Ler x Col 4x Replicate 1 H3K27me3 ChIP | 27,448,642 | 61.5 | 1,214,486 | 681,823 | |
| Ler x Col 4x Replicate 2 Input | 40,500,367 | 66.4 | 2,720,117 | 2,483,912 | 97.7 |
| Ler x Col 4x Replicate 2 H3 ChIP | 32,322,049 | 71.9 | 2,304,635 | 2,068,612 | |
| Ler x Col 4x Replicate 2 H3K27me3 ChIP | 34,978,215 | 63 | 2,681,981 | 1,636,717 | |

The purity of INTACT-extracted endosperm nuclei is indicated in the last column and was calculated as described in *Moreno-Romero et al. (2017)*.

## PHE1 regulates imprinted genes

Here, we report that many imprinted genes are targets of PHE1. Interestingly, we observed that binding of PHE1 to maternal and paternal alleles of some of these genes is conditioned by the epigenetic status of the alleles. The accumulation of DNA methylation near the TSS region has been reported to have a negative effect on gene expression (*Niederhuth et al., 2016*). In line with this, DNA methylation was shown to impair the binding of a wide range of plant TFs (*O'Malley et al., 2016*). Our observation that CG methylation restricts the binding of PHE1 to the paternal alleles of the tested MEGs is similar to observations made on the mammalian *Peg3* gene, where the TF YY1 shows methylation-dependent binding (*Kim, 2003*). Thus, our data reveal that in plants, as in animals, parental asymmetries of DNA methylation at *cis*-regulatory regions can lead to parent-of-origin-specific expression patterns (*Figure 7*).

Surprisingly, for PEG targets, PHE1 binding was detected not only on the expressed paternal allele, but also on the transcriptionally inactive maternal alleles, probably enabled by an H3K27me3-depleted island in these alleles. We speculate that this island could be important for PHE1-mediated recruitment of PRC2 complexes in the endosperm and, consequently, for the maintenance of H3K27me3 levels during its proliferation (*Figure 7*). This hypothesis is supported by the findings that plant PRC2 can be recruited by different families of TFs (*Xiao et al., 2017*; *Zhou et al., 2018*), and that the presence of *cis*-regulatory features such as Polycomb response elements is essential for maintenance of H3K27me3 upon cell division (*Coleman and Struhl, 2017*; *Laprell et al., 2017*). The recruitment of this epigenetic mark at imprinted loci takes place in the central cell, and is faithfully inherited in the endosperm after fertilization (*Gehring, 2013*; *Rodrigues and Zilberman, 2015*). Given that MADS-box DNA-binding motifs can be shared between different MADS-box TFs (*Aerts et al., 2018*), it is tempting to speculate that the PHE1 binding sites that are present at PEG loci could be shared by a central cell-specific MADS-box TF, and used to initiate PRC2 recruitment to these regions; nevertheless, this hypothesis remains to be tested formally.

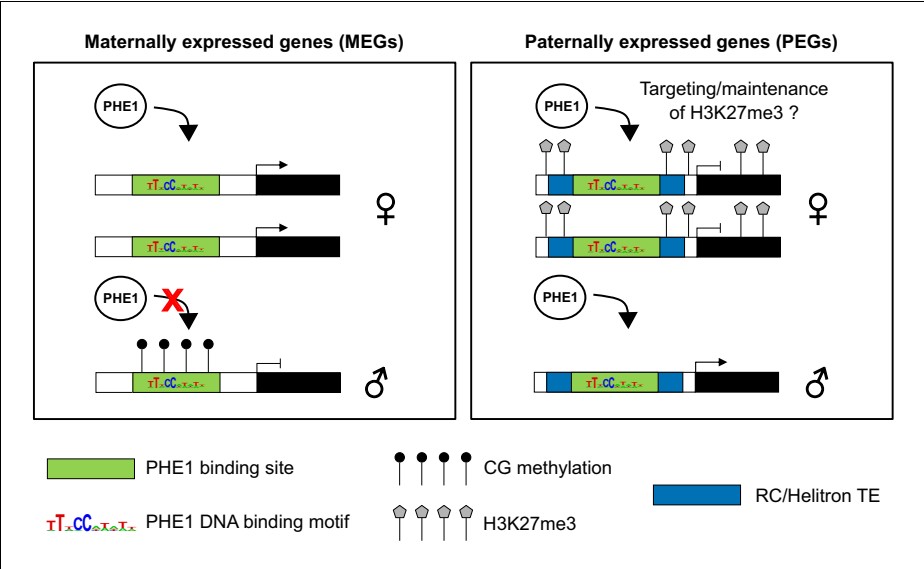

**Figure 7.** Control of imprinted gene expression by PHE1. Schematic model of imprinted gene control by PHE1. Maternally expressed genes (left panel) show DNA hypermethylation of paternal (♂) PHE1 binding sites. This precludes PHE1 accessibility to paternal alleles, leading to predominant binding and transcription from maternal (♀) alleles. In paternally expressed genes (right panel), RC/Helitrons found in flanking regions carry PHE1 DNA-binding motifs, allowing PHE1 binding. The paternal PHE1 binding site is devoid of repressive H3K27me3, facilitating the binding of PHE1 and transcription of this allele. H3K27me3 accumulates at the flanks of maternal PHE1 binding sites, whereas the binding sites remain devoid of this repressive mark. PHE1 is able to bind maternal alleles, but fails to induce transcription. We hypothesize that the accessibility of maternal PHE1 binding sites might be important for deposition of H3K27me3 during central cell development (possibly by another type I MADS-box transcription factor). It may also be required for the maintenance of H3K27me3 during endosperm proliferation.

## RC/Helitrons distribute PHE1 binding sites and generate novel transcriptional networks

Our work establishes a novel role for RC/Helitrons in the regulation of gene expression by showing that these TEs contain PHE1 binding sites. Our data favor a scenario in which these elements have been domesticated to function as providers of *cis*-regulatory sequences that facilitate transcription. Insertion of these elements can contribute to the generation of novel gene promoters that ensure the timely expression of nearby genes in the endosperm, controlled by PHE1 and possibly by other type I MADS-box TFs. We show that genes that are controlled by PHE1 are involved in key developmental pathways, such as endosperm proliferation and cellularization, with many of them being PEGs. Given our observation that RC/Helitrons carry PHE DNA-binding motifs, and that these TEs have been previously shown to be associated with PEGs (*Hatorangan et al., 2016*; *Pignatta et al., 2014*; *Wolff et al., 2011*), we propose a dual role for these elements in imprinting. Besides promoting the establishment of epigenetic modifications that are conducive to imprinting (*Gehring, 2013*; *Rodrigues and Zilberman, 2015*), they can also contribute to the transcriptional activation of these genes in the endosperm by carrying type I MADS-box TF binding sites. Future work will be required to validate this model and to evaluate the magnitude of its impact in the generation of imprinted expression across different plant species.

Binding sites for type II MADS-box TFs, as well as for E2F TFs, have been shown to be amplified by TEs (*Hénaff et al., 2014*; *Muiño et al., 2016*). These results, together with our study, provide examples of the TE-mediated distribution of TF binding sites throughout flowering plant genomes, adding support to the long-standing idea that transposition facilitates the formation of *cis*-regulatory architectures that are required to control complex biological processes (*Britten and Davidson, 1971*; *Feschotte, 2008*; *Hirsch and Springer, 2017*). We speculate that this process may have contributed to endosperm evolution by allowing the recruitment of crucial developmental genes into a

single transcriptional network, regulated by type I MADS-box TFs. The diversification of the mammalian placenta has been connected with the dispersal of hundreds of placenta-specific enhancers by endogenous retroviruses (*Chuong et al., 2013*; *Dunn-Fletcher et al., 2018*), suggesting that TE-mediated distribution of regulatory sequences has been relevant both for the evolution of the endosperm in flowering plants and for the evolution of the mammalian placenta.

In summary, this work reveals that the type I MADS-box TF PHE1 is a major regulator of imprinted genes and other genes required for endosperm development. The transcriptional control of these genes by PHE1 has been facilitated by RC/Helitron TEs, which amplify PHE1 DNA-binding sequences. Finally, we show how PHE1 can establish endosperm-based reproductive barriers, emphasizing a key role of type I MADS-box TFs in this process.

# Materials and methods

## Plant material and growth conditions

*Arabidopsis thaliana* seeds were sterilized in a closed vessel containing chlorine gas for 3 hr. Chlorine gas was produced by mixing 3 mL HCl 37% and 100 mL of 100% commercial bleach. Sterile seeds were plated in ½ MS-medium (0.43% MS salts, 0.8% Bacto Agar, 0.19% MES hydrate) supplemented with 1% sucrose. When required, the medium was supplemented with appropriate antibiotics. Seeds were stratified for 48 hr, at 4°C, in darkness. Plates containing stratified seeds were transferred to a long-day growth chamber (16 hr light/8 hr dark; 110 µmol s$^{-1}$m$^{-2}$; 21°C; 70% humidity), where seedlings grew for 10 days. After this period, the seedlings were transferred to soil and placed in a long-day growth chamber.

Several mutant lines used in this study have been previously described: *osd1-1* (*d'Erfurth et al., 2009*), *osd1-3* (*Heyman et al., 2011*) and *pi-1* (*Goto and Meyerowitz, 1994*). The *phe2* allele corresponds to a T-DNA insertion mutant (SALK_105945). Phenotypical analysis of this mutant revealed no deviant phenotype relative to Col wt plants (data not shown). Genotyping of *phe2* was done using the following primers (PHE2 fw 5'-AAATGTCTGGTTTTATGCCCC-3', PHE2 rv 5'-GTAGCGA-GACAATCGATTTCG-3', T-DNA 5'-ATTTTGCCGATTTCGGAAC-3').

## Generation of *phe1 phe2*

The *phe1 phe2* double mutant was generated using the CRISPR/Cas9 technique. A 20-nt sgRNA targeting *PHE1* was designed using the CRISPR Design Tool (*Ran et al., 2013*). A single-stranded DNA oligonucleotide corresponding to the sequence of the sgRNA, as well as its complementary oligonucleotide, was synthesized. BsaI restriction sites were added at the 5' and 3' ends, as represented by the underlined sequences (sgRNA fw 5'- ATTGCTCCTGGATCGAGTTGTAC-3'; sgRNA rv 5'-AAACGTACAACTCGATCCAGGAG-3'). These two oligonucleotides were then annealed to produce a double-stranded DNA molecule.

The double-stranded oligonucleotide was ligated into the egg-cell specific pHEE401E CRISPR/Cas9 vector (Z-P; *Wang et al., 2015b*) through the BsaI restriction sites. This vector was transformed into the *Agrobacterium tumefaciens* strain GV3101, and *phe2/–* plants were subsequently transformed using the floral-dip method (*Clough and Bent, 1998*).

To screen for T$_1$ mutant plants, we performed Sanger sequencing of *PHE1* amplicons that were derived from these plants and obtained with the following primers (fw 5'-AGTGAG-GAAAACAACATTCACCA-3'; rv 5'-GCATCCACAACAGTAGGAGC-3'). The selected mutant contained a homozygous 2-bp deletion that leads to a premature stop codon, and therefore to a truncated PHE1$_{1-50aa}$ protein. In the T$_2$ generation, the segregation of pHEE401E allowed the selection of plants that did not contain this vector and that were double homozygous *phe1 phe2* mutants. Genotyping of the *phe1* allele was done using primers fw 5'- AAGGAAGAAAGGGATGCTGA-3' and rv 5'-TCTGTTTCTTTGGCGATCCT-3'', followed by RsaI digestion.

## Seed imaging

Analysis of endosperm cellularization status was carried out by following the Feulgen staining protocol described previously (*Batista et al., 2019*). Imaging of Feulgen-stained seeds was done using a Zeiss LSM780 NLO multiphoton microscope, with excitation wavelength of 800 nm and acquisition between 520 nm and 695 nm.

## Chromatin immunoprecipitation (ChIP)

To find targets of PHE1, we performed ChIP using a *PHE1::PHE1–GFP* reporter line, which contains the *PHE1* promoter, its coding sequence and its 3′ regulatory sequence in the Col background (*Weinhofer et al., 2010*). Crosslinking of plant material was done by collecting 600 mg of 2 days after pollination (DAP) *PHE1::PHE1-GFP* siliques, and vacuum infiltrating them with a 1% formaldehyde solution in PBS. The vacuum infiltration was done for two periods of 15 min, with a vacuum release between each period. The crosslinking was then stopped by adding 0.125 mM glycine in PBS and performing a vacuum infiltration for a total of 15 min, with a vacuum release each 5 min. The material was then ground in liquid nitrogen, resuspended in 5 mL Honda buffer (*Moreno-Romero et al., 2017*), and incubated for 15 min with gentle rotation. This mixture was filtered twice through Miracloth and once through a CellTrics filter (30 μm), after which a centrifugation for 5 min, at 4°C and 1500 g was performed. The nuclei pellet was then resuspended in 100 μL of nuclei lysis buffer (*Moreno-Romero et al., 2017*), and the ChIP protocol was continued as described before (*Moreno-Romero et al., 2017*). ChIP DNA was isolated using the Pure Kit v2 (Diagenode), following the manufacturer's instructions.

For the parental-specific PHE1 ChIP, the starting material consisted of 600 mg of 2 DAP siliques from crosses between a *pi-1* maternal (L*er* accession) and a *PHE1::PHE1–GFP* paternal (Col accession) plant. The male sterile *pi-1* mutant was used to avoid emasculation of maternal plants. The crosslinking of plant material, nuclei isolation, ChIP protocol, and ChIP-DNA purification were the same as described before.

To assess parental-specific H3K27me3 profiles in 3x seeds, the INTACT system was used to isolate 4 DAP endosperm nuclei of seeds derived from L*er pi-1* x Col *INTACT osd1-1* crosses, as described previously (*Jiang et al., 2017*; *Moreno-Romero et al., 2017*). ChIPs against H3 and H3K27me3 were then performed on the isolated endosperm nuclei, following the previously described protocol (*Moreno-Romero et al., 2017*).

The antibodies used for these ChIP experiments were as follows: GFP Tag Antibody (A-11120, Thermo Fisher Scientific), anti-H3 (Sigma, H9289), and anti-H3K27me3 (Millipore, cat. no. 07–449). All experiments were performed with two biological replicates.

## RNA extraction

Seeds of 20 siliques of the crosses ♀ *pi-1* x ♂ wt, ♀ *pi-1* x ♂ *osd1-3*, and ♀ *pi-1* x ♂ *phe1 phe2 osd1-3* were harvested in RNA*later* (Invitrogen) at 6 DAP. For each cross, two biological replicates were generated. Total RNA was extracted with the *mir*Vana RNA isolation kit (Invitrogen), according to the manufacturer's instructions.

## Library preparation and sequencing

*PHE1::PHE1–GFP* ChIP libraries were prepared using the Ovation Ultralow v2 Library System (NuGEN), with a starting material of 1 ng, following the manufacturer's instructions. These libraries were sequenced at the SciLife Laboratory (Uppsala, Sweden), on an Illumina HiSeq2500 platform, using 50-bp single-end reads.

Library preparation and sequencing of H3K27me3 ChIPs in 3x seeds was performed as described previously (*Jiang et al., 2017*). mRNA libraries were prepared using the NEBNext Ultra II RNA Library Prep Kit For Illumina in combination with the NEBNext Poly(A) mRNA Magnetic Isolation Module, and the NEBNext Multiplex Oligos for Illumina. These libraries were sequenced at Novogene (Hong Kong), on an Illumina NovaSeq platform, using 150-bp paired-end reads.

All datasets were deposited at NCBI's Gene Expression Omnibus database (https://www.ncbi.nlm.nih.gov/geo/), under the accession number GSE129744.

## qPCR and Sanger sequencing of parental-specific PHE1 ChIP

Purified ChIP DNA and its respective input DNA obtained from the parental-specific PHE1 ChIP were used to perform qPCR. Positive and negative genomic regions for PHE1 binding were amplified using the following primers: AT1G55650 (fw 5′-CGAAGCGAAAAAGCACTCAC-3′; rv 5′-CCTTTTACATAATCCGCGTTAAA-3′), AT2G28890 (fw 5′-TTTGTGGTTGGAGGTTGTGA-3′; rv 5′-GTTGTTCGTGCCCATTTCTT-3′), AT5G04250 (fw 5′-AATTGACAAATGGTGTAATGGT-3′; rv 5′-CCAAAGAATTTGTTTTTCTATTCC-3′), AT1G72000 (fw 5′-AACAAATATGCACAAGAAGTGC-3′; rv 5′-ACC

TAGCAAGCTGGCAAAAC-3′), AT3G18550 (fw 5′-TCCTTTTCCAAATAAAGGCATAA-3′; rv 5′-AAA
TGAAAGAAATAAAAGGTAATGAGA-3′), AT2G20160 (fw 5′-TCCTAAATAAGGGAAGAGAAAGCA-
3′; rv 5′-TGTTAGGTGAAACTGAATCCAA-3′), negative region (fw 5′-TGGTTTTGCTGGTGATGATG-
3′, rv 5′-CCATGACACCAGTGTGCCTA-3′). HOT FIREPol EvaGreen qPCR Mix Plus (ROX) (Solis Bio-
dyne) was used as a master mix for qPCR amplification in a iQ5 qPCR system (Bio-Rad).

For Sanger sequencing, positive genomic regions for PHE1 binding containing SNPs that allowed
distinction between parents were amplified by PCR using the Phusion High-Fidelity DNA Polymerase
(Thermo Fisher Scientific), in combination with the primers described above. Amplified DNA was
purified using the GeneJET PCR Purification kit (Thermo Fisher Scientific) and used for Sanger
sequencing. The chromatograms obtained from Sanger sequencing were then analyzed for the pres-
ence of SNPs. The maternal:total read ratios were retrieved from the publications where these genes
were identified as imprinted.

## Bioinformatic analysis of ChIP-seq data

For the *PHE1::PHE1–GFP* ChIP, reads were aligned to the *Arabidopsis* (TAIR10) genome using Bow-
tie version 1.2.2 (*Langmead, 2010*), allowing two mismatches (-v 2). Only uniquely mapped reads
were kept. ChIP-seq peaks were called using MACS2 version 2.1.1, with its default settings (`-gsi-
ze 1.119e8, -bw 250`) (*Zhang et al., 2008*). Input samples served as control for their correspond-
ing GFP ChIP sample. Each biological replicate was handled individually using the same peak-calling
settings, and only the peak regions that overlap between the two replicates were considered for fur-
ther analysis. These regions are referred to throughout the text as PHE1 binding sites
(*Tables 1* and *2*). Peak overlap was determined with BEDtools version 2.26.0 (*Quinlan and Hall,
2010*). Each PHE1 binding site was annotated to a genomic feature and matched with a target gene
using the peak annotation feature (annotatePeaks.pl) provided in HOMER version 4.9 (*Heinz et al.,
2010*) (*Tables 1* and *2*). Only binding sites located less than 1.5 kb upstream to 0.5 kb downstream
of the nearest TSS were considered.

PHE1 DNA-binding motifs were identified from *PHE1::PHE1–GFP* ChIP-seq peak regions with
HOMER's findMotifsGenome.pl function, using the default settings. P-values of motif enrichment, as
well as alignments between PHE1 motifs and known motifs, were generated by HOMER.

Read mapping, coverage analysis, purity calculations, normalization of data, and determination of
parental origin of reads derived from H3K27me3 ChIPs in 3x seeds was performed following previ-
ously published methods (*Moreno-Romero et al., 2016*) (*Table 4*).

## Bioinformatic analysis of RNA-seq data

For each replicate, 150-bp-long reads were mapped to the *Arabidopsis* (TAIR10) genome, masked
for rRNA genes, using TopHat v2.1.0 (*Trapnell et al., 2009*) (parameters adjusted as –g 1 –a 10 –i
40 –I 5000 –F 0 –r 130). Gene expression was normalized to reads per kb per million mapped reads
(RPKM) using GFOLD (*Feng et al., 2012*). Expression level for each condition was calculated using
the mean of the expression values in both replicates. Genes that were regulated differentially across
the two replicates were detected using the rank product method, as implemented in the Bioconduc-
tor RankProd Package (*Hong et al., 2006*). Gene deregulation was assessed in the following combi-
nations: *osd1* vs. wt (♀ *pi-1* x ♂ *osd1-3* vs. ♀ *pi-1* x ♂ wt), *phe1 phe2 osd1* vs. wt (♀ *pi-1* x ♂ *phe1
phe2 osd1-3* vs. ♀ *pi-1* x ♂ wt), and *phe1 phe2 osd1* vs. *osd1* (♀ *pi-1* x ♂ *phe1 phe2 osd1-3* vs. ♀ *pi-
1* x ♂ *osd1-3*). Only genes with at least 10 reads in one of the conditions were considered, and a
pseudocount value of 10e–5 was added to genes that had no expression. Genes showing a
p-value <0.05, as determined by RankProd, were identified as deregulated.

## Analysis of PHE1 target genes

Significantly enriched Gene Ontology terms within target genes of PHE1 were identified using the
PLAZA 4.0 workbench (*Van Bel et al., 2018*), and further summarized using REVIGO (*Supek et al.,
2011*).

Enrichment of specific TF families within PHE1 targets was calculated by first normalizing the
number of PHE1-targeted TFs in each family, to the total number of TFs targeted by PHE1. As a con-
trol, the number of TFs belonging to a certain family was normalized to the total number of TFs in
the *Arabidopsis* genome. The Log$_2$ fold-change between these ratios was then calculated for each

family. The significance of the enrichment was assessed using the hypergeometric test. Annotation of TF families was carried out following the Plant Transcription Factor Database version 4.0 (*Jin et al., 2014*). Only TF families containing more than five members were considered in this analysis.

To determine which imprinted genes are targeted by PHE1, a custom list was used (*Figure 4—source data 1*). In this list, MEGs were considered to be those identified in *Schon and Nodine (2017)*, whereas PEGs were considered to include all of the genes identified in *Pignatta et al. (2014)*, *Schon and Nodine (2017)*, and *Wolff et al., 2011*.

To determine the proportion of genes overexpressed in paternal excess crosses that are targeted by PHE1, a previously published transcriptome dataset of 3x seeds was used (*Schatlowski et al., 2014*).

## Spatial overlap of TEs and PHE1 binding sites

Spatial overlap between PHE1 ChIP-seq peak regions (binding sites) and TEs was determined using the regioneR package version 1.8.1 (*Gel et al., 2015*), implemented in R version 3.4.1 (*R Development Core Team, 2017*). As a control, a mock set of binding sites was created, which we refer to as random binding sites. This random binding site set had the same total number of binding sites and the same size distribution as the PHE1 binding site set. Using regioneR, a Monte Carlo permutation test with 10,000 iterations was performed. In each iteration, the random binding sites were arbitrarily shuffled in the 3-kb promoter region of all *A. thaliana* genes. From this shuffling, the average overlap and standard deviation of the random binding site set was determined, as well as the statistical significance of the association between PHE1 binding sites and TE superfamilies/families.

BedTools version 2.26.0 (*Quinlan and Hall, 2010*) was used to determine the fraction of PHE1 binding sites targeting MEGs, PEGs, or non-imprinted genes where a spatial overlap between binding sites, RC/Helitrons and PHE1 DNA-binding motifs is simultaneously observed. The hypergeometric test was used to assess the significance of the enrichment of PHE1 binding sites where this overlap is observed, across different target types.

## Expression analysis of genes flanked by RC/Helitrons

We investigated the expression level of genes containing RC/Helitrons in their promoter regions (defined as 3 kb upstream of the TSS) in the embryo, endosperm, and seed coat of pre-globular stage seeds. Affymetrix GeneChip ATH1 Arabidopsis Genome Array data were extracted from *Belmonte et al. (2013)*. The expression values in micropylar endosperm, peripheral endosperm, and chalazal endosperm were averaged to represent the endosperm expression level. The endosperm expression levels of a given gene were then compared to the expression levels in the embryo and seed coat.

RC/Helitron-associated genes were classified into three groups: (i) genes with RC/Helitrons and PHE1 DNA-binding motifs within the 3 kb promoter, thus representing genes with domesticated RC/Helitrons; (ii) genes having RC/Helitrons and PHE1 DNA-binding motifs within the 3 kb promoter, but not bound by PHE1; and (iii) genes having RC/Helitrons located within the 3 kb promoter, and no PHE1 DNA-binding motifs. A two-tailed Mann-Whitney test with continuity correction was used to assess the statistical significance of differences in expression levels between gene groups.

## Calculation of PHE1 DNA-binding motif densities

To measure the density of PHE1 DNA-binding motifs within different genomic regions of interest, the fasta sequences of these regions were first obtained using BEDtools. HOMER's scanMotifGenomeWide.pl function was then used to screen these sequences for the presence of PHE1 DNA-binding motifs, and to count the number of occurrences of each motif. Motif density was then calculated as the number of occurrences of each motif, normalized to the size of the genomic region of interest. Motif densities in RC/Helitron consensus sequences were calculated as above. Perfect PHE1 DNA-binding motif sequences were defined as those represented in *Figure 3a*. Nearly perfect motif sequences were defined as sequences in which only one nucleotide substitution could give rise to a perfect PHE1 DNA-binding motif. Consensus sequences were obtained from Repbase (*Bao et al., 2015*). Chi-square tests of independence were used to test whether there were any associations

between specific genomic regions and PHE1 DNA-binding motifs. This was done by comparing the proportion of DNA bases corresponding to PHE1 DNA-binding motifs in each genomic region.

## Identification of homologous PHE1 DNA-binding motifs carried by RC/Helitrons

To assess the homology of PHE1 DNA-binding motifs and associated RC/Helitron sequences, pairwise comparisons were made among all sequences, using the BLASTN program. The following parameters were followed: word size = 7, match/mismatch scores = 2/–3, gap penalties, existence = 5, extension = 2. The RC/Helitron sequences were considered to be homologous if the alignment covered at least 9 bp out of the 10-bp PHE1 DNA-binding motif sites, extended longer than 30 bp, and had more than 70% identity. Because the mean length of intragenomic conserved non-coding sequences is around 30 bp in *A. thaliana* (*Thomas et al., 2007*), we considered this as the minimal length of alignments needed to define a pair of related motif-carrying TE sequences. The pairwise homologous sequences were then merged into higher-order clusters, on the basis of shared elements in the homologous pairs.

## Phylogenetic analyses of PHE1-targeted PEG orthologs in the Brassicaceae

Amino-acid sequences and nucleotide sequences of PHE1-targeted PEGs were obtained from TAIR10. The sequences of homologous genes in the Brassicaceae and several other rosids were obtained in PLAZA 4.0 (https://bioinformatics.psb.ugent.be/plaza/) (*Van Bel et al., 2018*), BRAD database (http://brassicadb.org/brad/) (*Wang et al., 2015a*), and Phytozome v.12 (https://phytozome.jgi.doe.gov/) (*Goodstein et al., 2012*).

For each PEG of interest, the amino-acid sequences of the gene family were used to generate a guided codon alignment by MUSCLE with default settings (*Edgar, 2004*). A maximum likelihood tree was then generated by IQ-TREE 1.6.7 with codon alignment as the input (*Nguyen et al., 2015*). The implemented ModelFinder was executed to determine the best substitution model (*Kalyaanamoorthy et al., 2017*), and 1000 replicates of ultrafast bootstrap were applied to evaluate the branch support (*Hoang et al., 2018*). The tree topology and branch supports were reciprocally compared with, and supported by, another maximum likelihood tree generated using RAxML v. 8.1.2 (*Stamatakis, 2014*).

We selected PEGs that had well-supported gene family phylogeny with no lineage-specific duplication in the *Arabidopsis* and *Capsella* clades, and where imprinting data were available for all *Capsella grandiflora* (Cgr), *C. rubella* (Cru), and *Arabidopsis lyrata* (Aly) orthologs of interest (*Klosinska et al., 2016*; *Lafon-Placette et al., 2018*; *Pignatta et al., 2014*). We then obtained the promoter region, defined as 3 kb upstream of the TSS, of the orthologs and paralogs in Brassicaceae and rosids species. These promoter sequences were searched for the presence of homologous RC/Helitron sequences, as well as for putative PHE1 DNA-binding sites contained in these TEs.

Homology between *A. thaliana* RC/Helitron sequences and Brassicaceae sequences was detected by aligning the *A. thaliana* sequence to the promoter region of the orthologous PEG, using the BLASTN program, with the following parameters: word size = 11, match/mismatch scores = 2/–3, gap penalties, existence = 5, extension = 2. Aligned sequences are considered homologous if they spanned more than 100 nt of the *A, thaliana* RC/Helitron with over 60% identity.

## Epigenetic profiling of PHE1 binding sites

Parental-specific H3K27me3 profiles (*Moreno-Romero et al., 2016*) and DNA methylation profiles (*Schatlowski et al., 2014*) generated from endosperm of 2x seeds were used for this analysis. Levels of H3K27me3 and CG DNA methylation were quantified in each 50-bp bin across the 2-kb region surrounding PHE1 binding site centers using deepTools version 2.0 (*Ramírez et al., 2016*). These values were then used to generate H3K27me3 heatmaps and metagene plots, as well as boxplots of CG methylation in PHE1binding sites. Clustering analysis of H3K27me3 distribution in PHE1-binding sites was done following the k-means algorithm as implemented by deepTools. A two-tailed Mann-Whitney test with continuity correction was used to assess statistical significance of differences in CG methylation levels.

## Parental gene expression ratios in 2x and 3x seeds

To determine parental gene expression ratios in 2x and 3x seeds, we used previously generated endosperm gene expression data (*Martinez et al., 2018*). In this dataset, L*er* plants were used as maternal plants pollinated with wt Col or *osd1* Col plants, allowing determination of the parental origin of sequenced reads following the method described before (*Moreno-Romero et al., 2019*).

Parental gene expression ratios were calculated as the number of maternally derived reads divided by the sum of maternally and paternally derived reads available for any given gene. Ratios were calculated separately for the two biological replicates of each cross (L*er* x wt Col and L*er* x *osd1* Col), and the average of both replicates was considered for further analysis. The MEG and PEG ratio thresholds for 2x and 3x seeds indicated in *Figure 6d* were defined as a four-fold deviation of the expected read ratios, towards more maternal or paternal read accumulation, respectively. The expected read ratio for a biallelically expressed gene in 2x seeds is two maternal reads:three total reads, whereas for 3x paternal excess seeds this ratio is two maternal reads:four total reads. Deviations from these expected ratios were used to classify the expression of published imprinted genes (*Figure 4—source data 1*) as maternally or paternally biased in 3x seeds, according to the direction of the deviation. As a control, the parental bias of these imprinted genes was also assessed in 2x seeds.

## Parental expression ratios of genes associated with H3K27me3 clusters

Previously published endosperm gene expression data, generated with the INTACT system, were used for this analysis (*Del Toro-De León and Köhler, 2019*). Parental gene expression ratios were determined as the mean between ratios observed in the L*er* x Col cross and its reciprocal cross. As a reference, the parental gene expression ratio for all endosperm expressed genes was also determined. A two-tailed Mann-Whitney test with continuity correction was used to assess statistical significance of differences between parental gene expression ratios.

## H3K27me3 accumulation in imprinted genes

Parental-specific accumulation of H3K27me3 across imprinted gene bodies in the endosperm of 2x (*Moreno-Romero et al., 2016*) and 3x seeds (this study) was estimated by calculating the mean values of the H3K27me3 z-score across the gene length. Imprinted genes were considered as those genes previously identified in different studies (*Gehring et al., 2011*; *Hsieh et al., 2011*; *Pignatta et al., 2014*; *Schon and Nodine, 2017*; *Wolff et al., 2011*). A two-tailed Mann-Whitney test with continuity correction was used to assess statistical significance of differences in H3K27me3 z-score levels.

## Statistics

Sample size, statistical tests used, and respective p-values are indicated in each figure or figure legend, and further specified in the corresponding Methods sub-section.

## Data availability

ChIP-seq and RNA-seq data generated in this study are available at NCBI's Gene Expression Omnibus database (https://www.ncbi.nlm.nih.gov/geo/), under the accession number GSE129744. Additional data used to support the findings of this study are available at NCBI's Gene Expression Omnibus, under the following accession numbers: H3K27me3 ChIP-seq data from 2x endosperm (*Moreno-Romero et al., 2016*) – GSE66585; gene expression data in 2x and 3x endosperm (*Martinez et al., 2018*) – GSE84122; gene expression data in 2x and 3x seeds and parental-specific DNA methylation from 2x endosperm (*Schatlowski et al., 2014*) – GSE53642; parental-specific gene expression data of 2x INTACT-isolated endosperm nuclei (*Del Toro-De León and Köhler, 2019*) – GSE119915. Gene expression profile of different seed compartments (*Belmonte et al., 2013*) – GSE12404.

## Materials and correspondence

The materials generated in this study are available upon request to CK (claudia.kohler@slu.se).

## Acknowledgements
We thank Qi-Jun Chen for providing the pHEE401E CRISPR/Cas9 vector. We are grateful to Cecilia Wärdig for technical assistance. Sequencing was performed by the SNP and SEQ Technology Platform, Science for Life Laboratory at Uppsala University, a national infrastructure supported by the Swedish Research Council (VRRFI) and the Knut and Alice Wallenberg Foundation. This research was supported by a grant from the Swedish Research Council (to CK), a grant from the Knut and Alice Wallenberg Foundation (to CK), and support from the Göran Gustafsson Foundation for Research in Natural Sciences and Medicine (to CK).

## Additional information

### Funding

| Funder | Author |
| --- | --- |
| Vetenskapsrådet | Claudia Köhler |
| Knut och Alice Wallenbergs Stiftelse | Claudia Köhler |
| Göran Gustafsson's Foundation for Science and Medical Research | Claudia Köhler |

The funders had no role in study design, data collection and interpretation, or the decision to submit the work for publication.

### Author contributions
Rita A Batista, Conceptualization, Data curation, Software, Formal analysis, Validation, Investigation, Visualization, Methodology, Project administration; Jordi Moreno-Romero, Conceptualization, Data curation, Formal analysis, Supervision, Validation, Investigation, Visualization, Methodology, Project administration; Yichun Qiu, Conceptualization, Resources, Data curation, Formal analysis, Supervision, Funding acquisition, Validation, Investigation, Methodology, Project administration; Joram van Boven, Conceptualization, Data curation, Formal analysis, Validation, Investigation, Visualization, Methodology; Juan Santos-González, Conceptualization, Data curation, Software, Formal analysis, Validation, Investigation, Visualization, Methodology; Duarte D Figueiredo, Conceptualization, Data curation, Formal analysis, Supervision, Validation, Investigation, Project administration; Claudia Köhler, Conceptualization, Resources, Data curation, Software, Formal analysis, Supervision, Funding acquisition, Validation, Visualization, Methodology, Project administration

### Author ORCIDs
Rita A Batista (iD) https://orcid.org/0000-0002-2083-4622
Duarte D Figueiredo (iD) http://orcid.org/0000-0001-7990-3592
Claudia Köhler (iD) https://orcid.org/0000-0002-2619-4857

### Decision letter and Author response
Decision letter https://doi.org/10.7554/eLife.50541.sa1
Author response https://doi.org/10.7554/eLife.50541.sa2

## Additional files
### Supplementary files
• Transparent reporting form

### Data availability
ChIP-seq and RNA-seq data generated in this study is available at NCBI's Gene Expression Omnibus database, under the accession number GSE129744.

The following dataset was generated:

| Author(s) | Year | Dataset title | Dataset URL | Database and Identifier |
|---|---|---|---|---|
| Batista RA, Moreno-Romero J, van Boven J, Qiu Y, Santos-González J, Figueiredo DD, Köhler C | 2019 | The MADS-box transcription factor PHERES1 controls imprinting in the endosperm by binding to domesticated transposons | https://www.ncbi.nlm.nih.gov/geo/query/acc.cgi?acc=GSE129744 | NCBI Gene Expression Omnibus, GSE129744 |

The following previously published datasets were used:

| Author(s) | Year | Dataset title | Dataset URL | Database and Identifier |
|---|---|---|---|---|
| Moreno-Romero J, Jiang H, Santos-González J, Köhler C | 2015 | Parental epigenetic asymmetry of PRC2-mediated histone modifications in the Arabidopsis endosperm | https://www.ncbi.nlm.nih.gov/geo/query/acc.cgi?acc=GSE66585 | NCBI Gene Expression Omnibus, GSE66585 |
| Martinez G, Wolff P, Wang Z, Moreno-Romero J, Santos-Gonzalez J, Liu Conze L, DeFraia C, Slotkin K, Köhler C | 2016 | Paternal easiRNAs establish the triploid block in Arabidopsis | https://www.ncbi.nlm.nih.gov/geo/query/acc.cgi?acc=GSE84122 | NCBI Gene Expression Omnibus, GSE84122 |
| Santos-González JC, Köhler C | 2013 | DNA hypomethylation bypasses the interploidy hybridization barrier in Arabidopsis | https://www.ncbi.nlm.nih.gov/geo/query/acc.cgi?acc=GSE53642 | NCBI Gene Expression Omnibus, GSE53642 |
| Moreno-Romero J, Del Toro-De León G, Yadav VK, Santos-González J, Köhler C | 2018 | Epigenetic signatures associated with paternally-expressed imprinted genes in the endosperm | https://www.ncbi.nlm.nih.gov/geo/query/acc.cgi?acc=GSE119915 | NCBI Gene Expression Omnibus, GSE119915 |
| Belmonte MF, Kirkbride RC, Stone SL, Pelletier JM | 2008 | Expression data from Arabidopsis Seed Compartments at 5 discrete stages of development | https://www.ncbi.nlm.nih.gov/geo/query/acc.cgi?acc=GSE12404 | NCBI Gene Expression Omnibus, GSE12404 |

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
