## [Decision Letter]

**Acceptance summary:**

This paper elucidates the mechanism and evolution of imprinted gene regulation in the endosperm by examining the binding pattern of the type I MADS-box transcription factor PHE1 in Arabidopsis seeds and the role of PHE1 in mediating triploid block, the phenomenon where pollination with diploid pollen causes endosperm proliferation defects and seed abortion. PHE1 preferentially binds near transcription factor genes, especially of the type I MADS-box family, and imprinted genes – those preferentially expressed from either the maternal allele (MEGs) or paternal allele (PEGs). PHE1 binding sites are also frequently associated with Helitron TEs. DNA methylation of the paternal allele prevents PHE1 binding at MEGs, restricting binding to the maternal allele. PHE1 binding is associated with genes that are overexpressed due to triploid block and mutation of PHE1 rescues triploid block-induced seed abortion and reduces the expression of some overexpressed genes. Helitron transposition is proposed to have created a network of PHE1 regulated genes, including imprinted genes, that regulate endosperm development. This paper establishes PHE1 as an important regulator of imprinted genes and endosperm development.

**Decision letter after peer review:**

Thank you for submitting your article "The MADS-box transcription factor PHERES1 controls imprinting in the endosperm by binding to domesticated transposons" for consideration by *eLife*. Your article has been reviewed by three peer reviewers, one of whom is a member of our Board of Reviewing Editors, and the evaluation has been overseen by Christian Hardtke as the Senior Editor. The reviewers have opted to remain anonymous.

The reviewers have discussed the reviews with one another and the Reviewing Editor has drafted this decision to help you prepare a revised submission.

Essential revisions:

This is an important and interesting paper that elucidates the mechanism and evolution of imprinted gene regulation in the endosperm. However, the reviewers were concerned that the paper's core claim, that PHE1 "is a master regulator of paternally expressed imprinted genes, as well as of non‐imprinted key regulators of endosperm development" is not strongly supported. Several aspects of the analysis and data presentation should be improved and clarified, as outlined below. The most important revisions relate to points 6-9, as these directly influence the paper's main conclusions. In particular, all reviewers requested the inclusion of RNA-seq data that would evaluate how PHE1 affects transcription (point 7).

This paper addresses a complex topic and a great deal of important data (and the model) are relegated to supplementary figures. We feel that the paper would benefit from a less compressed presentation of the results and a separate Discussion section and should therefore be recast as a Research Article.

Presentation of ChIP-seq data and association with Helitron TEs:

1) PHE1 ChIP-seq data are central to this paper but are described only very briefly in Supplementary file 1. This table should be included as a supplement to Figure 1 (a similar table for H3K27me3 should also be included as a supplement to Figure 4), and other data and analyses would be helpful. Is PHE1 binding enriched near transcriptional start sites (TSSs), as for other plant TEs? What do PHE1 binding peaks look like near important genes examined in this paper?

2) It is not clear if the motifs in Figure 1A occur separately or are found together. Also, are the authors proposing that both are CArG-boxes? Only motif 2 looks like a bona fide CArG-box: CC(A/T)_6_GG.

3) It is not clear if there is an orientation to the meta-gene plot in Figure 2C with respect to the associated gene annotations. From the graph, it looks like H3K27me3 is accumulating 5' and 3' of the PHE1 binding site, but it would be informative to show the alignment according to the associated gene orientation. The figure appears to show that PHE1 binding takes place in a H3K27me3-depleted island within a H3K27me3-dense region (5' and 3'), which is also shown in the schematic model in Figure 2—figure supplement 3. Is this correct?

4) Related to the above, the authors state that "Interestingly, we observed biallelic binding in PEG targets (Figure 2E). Even though the maternal PHE1 binding sites in PEGs were flanked by H3K27me3 (Figure 2B-C), correlating with transcriptional repression of maternal alleles, the absence of this mark within the binding site centres seems to be permissive for maternal PHE1 binding." This is indeed unexpected and very interesting, and this point would be more convincing if the authors examine additional loci to confirm the biallelic binding. Have the authors performed ChIP-seq on the crosses used in Figure 2E?

5) In the expression profile in Figure 1D, the inclusion of PHE1 targets without flanking TEs would be helpful to judge the extent to which the presence of the TE impacts the expression level of the nearby gene.

Definition of PHE1 target genes and the effects of PHE1 on transcription:

6) The concept of a "PHE1 target" is central to the paper. According to the Materials and methods, PHE1 targets are genes with "binding sites located less than 3 kb away from the […] transcription start site" – genes with a PHE1 peak within a 6 kb window around the TSS. Most functional TF binding sites are much closer to the TSS. For example, conserved *Arabidopsis* TF binding sites peak sharply at -50 bp from the TSS and are not significantly enriched outside a 600 bp window (-400 bp to +200 bp) around the TSS (Yu et al., 2016). ChIP-seq data for *Arabidopsis* TFs shows similar patterns. Unless PHE1 is very different, expanding the window 10-fold will introduce a great deal of false-positive noise. This will not fundamentally affect conclusions based on analyses of large numbers of PHE1 "targets", but it will affect the validity of designating specific genes or groups of genes (as in 12% of MEGs, 31% of PEGs, 50% of highly upregulated genes in 3x seeds) as PHE1 targets. The authors should evaluate how a more conservative designation of PHE1 target affects their results and conclusions. Because this is so important, how PHE1 targets are designated should also be described in the Results.

7) The authors claim that PHE1 "is a master regulator of paternally expressed imprinted genes, as well as of non‐imprinted key regulators of endosperm development." However, this claim is based mostly on PHE1 binding data. In the vast majority of cases, the authors do not know that PHE1 regulates any of its target genes. Considering the permissive approach for designating genes as PHE1 targets described above, the lack of functional data for most genes means that the authors' claim is not strongly supported. The authors' claims would be much stronger if they could designate PHE1 targets based on transcriptional activation (or repression) as well as PHE1 binding. This would also allow the authors to evaluate if PHE1 is generally a transcriptional activator, and if PHE1 binding further from a gene (outside a putative promoter) is associated with transcriptional effects. RNA-seq of the RNA samples used in Figure 4—figure supplement 1F (wt x wt, wt x *osd1*, wt x *phe1 phe2*, wt x *phe1 phe2 osd1*) would allow a much better designation of PHE1 targets when cross-analysed with the ChIP-seq data.

Association with imprinted genes:

8) According to the Materials and methods, "To determine which imprinted genes are targeted by PHE1, a custom list consisting of the sum of imprinted genes identified in different studies was used (Figure 1—source data 1) (Gehring et al., 2011; Hsieh et al., 2011; Pignatta et al., 2014; Schon and Nodine, 2017; Wolff et al., 2011)." Schon and Nodine determined that earlier published lists of imprinted genes were substantially affected by maternal contamination and came up with high-confidence lists of MEGs and PEGs. Why did the authors not use these presumably more robust (but smaller) lists? It is concerning that in analyses comparing PHE1 targets with imprinted genes, both datasets likely contain many false-positives.

9) The significant enrichment of PEGs and MEGs noted in the fifth paragraph of the Results and Discussion section is based on the comparison with the total number of genes in the genome. A better reference set would be genes expressed in the endosperm (for instance: Belmonte et al., 2013), which would reduce the control set to approximately 12,000 instead of 27,400 genes.

10) The presence of RC/Helitrons in orthologs is very interesting and should be evaluated on a meta-genomic scale in addition to showing a few examples. Is this really an evolutionary conserved phenomenon and to which extent does it correlate with imprinting? Addressing this question would significantly strengthen this work, as the implication of the shown examples for the evolution of imprinting and gene regulation via Helitron TE insertion is not clear.

---

## [Author Response]

Essential revisions:This is an important and interesting paper that elucidates the mechanism and evolution of imprinted gene regulation in the endosperm. However, the reviewers were concerned that the paper's core claim, that PHE1 "is a master regulator of paternally expressed imprinted genes, as well as of non‐imprinted key regulators of endosperm development" is not strongly supported. Several aspects of the analysis and data presentation should be improved and clarified, as outlined below. The most important revisions relate to points 6-9, as these directly influence the paper's main conclusions. In particular, all reviewers requested the inclusion of RNA-seq data that would evaluate how PHE1 affects transcription (point 7).This paper addresses a complex topic and a great deal of important data (and the model) are relegated to supplementary figures. We feel that the paper would benefit from a less compressed presentation of the results and a separate Discussion section and should therefore be recast as a Research Article.

We thank the editors and reviewers for positive assessment of our work and the suggestions for improvement. As suggested, we have expanded the paper to a Research Article, and moved some of the supplementary figures to main figures. We have also addressed the reviewers’ comments as detailed below.

Presentation of ChIP-seq data and association with Helitron TEs:1) PHE1 ChIP-seq data are central to this paper but are described only very briefly in Supplementary file 1. This table should be included as a supplement to Figure 1 (a similar table for H3K27me3 should also be included as a supplement to Figure 4), and other data and analyses would be helpful. Is PHE1 binding enriched near transcriptional start sites (TSSs), as for other plant TEs? What do PHE1 binding peaks look like near important genes examined in this paper?

We have included a new Results section describing the ChIP-seq data, and the new Figure 1A shows the distribution of PHE1 binding sites across TSSs (subsection “Identification of PHE1 target genes”, first paragraph). We have also added information on binding sites, binding motifs, and TEs for three selected gene in Figure 1—figure supplement 2. Information on the PHE1 ChIP-seq mapping, peak calling, and annotation statistics can now be found in Figure 1—figure supplement 1.

2) It is not clear if the motifs in Figure 1A occur separately or are found together. Also, are the authors proposing that both are CArG-boxes? Only motif 2 looks like a bona fide CArG-box: CC(A/T)_6_GG.

A detailed analysis of frequency and co-occurrence of motifs is now included in Figure 3—figure supplement 1, and a discussion on the resemblance of PHE1 motifs and bona fide CArG motifs has been included in the Results (subsection “Transposable elements act as cis-regulatory elements by carrying PHE1 DNA-binding motifs”, first paragraph).

3) It is not clear if there is an orientation to the meta-gene plot in Figure 2C with respect to the associated gene annotations. From the graph, it looks like H3K27me3 is accumulating 5' and 3' of the PHE1 binding site, but it would be informative to show the alignment according to the associated gene orientation. The figure appears to show that PHE1 binding takes place in a H3K27me3-depleted island within a H3K27me3-dense region (5' and 3'), which is also shown in the schematic model in Figure 2—figure supplement 3. Is this correct?

Directional information of H3K27me3 distribution is now shown in Figure 4—figure supplement 1. This analysis reveals that PHE1 indeed binds in an H3K27me3-depleted island, irrespectively of the directionality of the target gene.

4) Related to the above, the authors state that "Interestingly, we observed biallelic binding in PEG targets (Figure 2E). Even though the maternal PHE1 binding sites in PEGs were flanked by H3K27me3 (Figure 2B-C), correlating with transcriptional repression of maternal alleles, the absence of this mark within the binding site centres seems to be permissive for maternal PHE1 binding." This is indeed unexpected and very interesting, and this point would be more convincing if the authors examine additional loci to confirm the biallelic binding. Have the authors performed ChIP-seq on the crosses used in Figure 2E?

So far, our attempts to do parental-specific PHE1 ChIP-seq were not successful. While agreeably interesting, we think that the time and resources required to generate this data goes beyond the scope of this manuscript. Nevertheless, we provide a few case-examples of Sanger-sequenced binding sites, and phrased the discussion to indicate the hypothesis derived from these observations are speculative and require additional experimental validation (subsection “PHE1 regulates imprinted genes”).

5) In the expression profile in Figure 1D, the inclusion of PHE1 targets without flanking TEs would be helpful to judge the extent to which the presence of the TE impacts the expression level of the nearby gene.

We have now included this control set in Figure 3D.

Definition of PHE1 target genes and the effects of PHE1 on transcription:6) The concept of a "PHE1 target" is central to the paper. According to the Materials and methods, PHE1 targets are genes with "binding sites located less than 3 kb away from the […] transcription start site" – genes with a PHE1 peak within a 6 kb window around the TSS. Most functional TF binding sites are much closer to the TSS. For example, conserved Arabidopsis TF binding sites peak sharply at -50 bp from the TSS and are not significantly enriched outside a 600 bp window (-400 bp to +200 bp) around the TSS (Yu et al., 2016). ChIP-seq data for Arabidopsis TFs shows similar patterns. Unless PHE1 is very different, expanding the window 10-fold will introduce a great deal of false-positive noise. This will not fundamentally affect conclusions based on analyses of large numbers of PHE1 "targets", but it will affect the validity of designating specific genes or groups of genes (as in 12% of MEGs, 31% of PEGs, 50% of highly upregulated genes in 3x seeds) as PHE1 targets. The authors should evaluate how a more conservative designation of PHE1 target affects their results and conclusions. Because this is so important, how PHE1 targets are designated should also be described in the Results.

We have now included an evaluation of the distribution of PHE1 binding sites in Figure 1A, and observed that it is similar to what has been observed for other *Arabidopsis* TFs (Yu et al., 2016).We have also narrowed down the promoter window for target gene identification, and only considered genes for which the binding sites were within -1.5 <-> -0.5 kb around the TSS. We believe this is justified given the peak distribution discussed above, and should not result in identification of false positives. Accordingly, we have revised all the results presented in the paper in order to reflect the new target gene list. Furthermore, we have included a Results section clarifying how PHE1 targets were identified (subsection “Identification of PHE1 target genes”, first paragraph).

7) The authors claim that PHE1 "is a master regulator of paternally expressed imprinted genes, as well as of non‐imprinted key regulators of endosperm development." However, this claim is based mostly on PHE1 binding data. In the vast majority of cases, the authors do not know that PHE1 regulates any of its target genes. Considering the permissive approach for designating genes as PHE1 targets described above, the lack of functional data for most genes means that the authors' claim is not strongly supported. The authors' claims would be much stronger if they could designate PHE1 targets based on transcriptional activation (or repression) as well as PHE1 binding. This would also allow the authors to evaluate if PHE1 is generally a transcriptional activator, and if PHE1 binding further from a gene (outside a putative promoter) is associated with transcriptional effects. RNA-seq of the RNA samples used in Figure 4—figure supplement 1F (wt x wt, wt x osd1, wt x phe1 phe2, wt x phe1 phe2 osd1) would allow a much better designation of PHE1 targets when cross-analysed with the ChIP-seq data.

We have generated transcriptomic data of wt x wt, wt x *osd1* and wt x *phe1 phe2 osd1* seeds. The results are now included in Figure 1D-E, and presented in the last paragraph of the subsection “Identification of PHE1 target genes”. We found a significant overlap between PHE1 targets and upregulated genes in *osd1* vs. wt, as well as between PHE1 targets and downregulated genes in *phe1 phe2 osd1* vs. *osd1* (Figure 1D). Nevertheless, not all of the PHE1 targets that were upregulated in *osd1 vs.* wt were significantly downregulated in *phe1 phe2 osd1* vs. *osd1*. One likely explanation is that PHE1 acts redundantly with several other AGLs, in addition to PHE2. In Figure 3—figure supplement 2 we show expression of *PHE1* paralogs, revealing that they remain highly upregulated in *phe1 phe2 osd1* vs. *osd1.* Redundancy likely also explains the incomplete rescue of triploid seeds by loss of PHE1 and PHE2 (as can be observed in Figure 6B-C).

Furthermore, while there is a significant overlap between PHE1 targets and genes being upregulated in *osd1* vs. wt, not all PHE1 targets are upregulated (Figure 1D). This can be explained by the fact that the time-points for the PHE1 ChIP-seq and for the transcriptome profiles differ (2DAP vs. 6DAP). The reason we used a later time-point for the transcriptome data is that we observe differences in *PHE1* expression between diploid and triploid seeds only at later stages of seed development (Erilova et al., 2009).

We have nevertheless observed that a considerable proportion of target genes shows an expression pattern across seed development that mimics the expression of PHE1 (Figure 1B-C), and is upregulated upon PHE1 overexpression as discussed above (Figure 1D), suggesting that PHE1 acts as a transcriptional activator.

We believe the analysis of expression of PHE1 targets across seed development, as well as in seeds where PHE1 is overexpressed (*osd1* vs. wt) or absent (*phe1 phe2 osd1* vs. osd) gives important insights into the role of PHE1 as a transcriptional regulator, and validates the targets obtained through ChIP-seq. Notwithstanding, and because of all the reasons discussed above, we refrained from integrating gene expression datasets into the pipeline for PHE1 target identification, as that would prevent the identification of biologically relevant genes.

Association with imprinted genes:8) According to the Materials and methods, "To determine which imprinted genes are targeted by PHE1, a custom list consisting of the sum of imprinted genes identified in different studies was used (Figure 1—source data 1) (Gehring et al., 2011; Hsieh et al., 2011; Pignatta et al., 2014; Schon and Nodine, 2017; Wolff et al., 2011)." Schon and Nodine determined that earlier published lists of imprinted genes were substantially affected by maternal contamination and came up with high-confidence lists of MEGs and PEGs. Why did the authors not use these presumably more robust (but smaller) lists? It is concerning that in analyses comparing PHE1 targets with imprinted genes, both datasets likely contain many false-positives.

For MEGs we used the genes identified in Schon and Nodine, 2017; while for PEGs we used the sum of genes identified by Gehring et al., 2011, Hsieh et al., 2011, and Pignatta et al., 2014. Maternal seed coat contamination does not bear the risk of false positives in PEG identification (Schon and Nodine, 2017) thus, we consider this approach as justified. We have clarified the data sources in the Materials and methods and in Figure 4—source data 1.

9) The significant enrichment of PEGs and MEGs noted in the fifth paragraph of the Results and Discussion section is based on the comparison with the total number of genes in the genome. A better reference set would be genes expressed in the endosperm (for instance: Belmonte et al., 2013), which would reduce the control set to approximately 12,000 instead of 27,400 genes.

Using only genes reported to be endosperm expressed in the Belmonte dataset restricts the total number of imprinted genes that can be evaluated, because many of these genes are not considered to be expressed in this dataset (see Author response image 1 versus Figure 4A). For this reason, we prefer to use all genes as control, as this provides a more representative picture of the imprinted gene set regulated by PHE1.

10) The presence of RC/Helitrons in orthologs is very interesting and should be evaluated on a meta-genomic scale in addition to showing a few examples. Is this really an evolutionary conserved phenomenon and to which extent does it correlate with imprinting? Addressing this question would significantly strengthen this work, as the implication of the shown examples for the evolution of imprinting and gene regulation via Helitron TE insertion is not clear.

Our previous study (Hatorangan et al., 2016) revealed that the accumulation of RC/Helitrons and MuDR elements in the vicinity of imprinted genes is a conserved feature in *Arabidopsis thaliana* and *Capsella rubella*. In this study, we further characterized the role of RC/Helitrons as the vectors of PHE1 binding sites. However, our ChIP-seq data revealed that not all genes with a PHE1 DNA-binding motif are targeted by PHE1: PHE1 DNA-binding motifs are highly abundant in both *Arabidopsis* and *Capsella* promoter regions (1kb upstream of TSS) (see Author response table 1, Author response table 2, Author response table 3). Therefore, similar ChIP-seq experiments are required in other Brassicaceae species in order to identify targets of PHE1 homologs, since predicting targets based on the presence of motifs is not possible. Nevertheless, PHE1 DNA-binding motifs were significantly enriched in *Arabidopsis* and *Capsella* PEG promoters, when compared to random sets of genes (see Author response table 1, Author response table 2, Author response table 3), supporting the notion that the presence of these motifs connects to imprinted expression. However, due to the poor annotation of RC/Helitrons in other species, testing whether these motifs are enriched within RC/Helitrons and how that relates to imprinting, would require an in-depth analysis that we believe goes beyond the scope of this manuscript.

**Author response table 1. resptable1:** 

*Capsella rubella*
	total number of genes	number of genes with PHE1 DNA-binding motifs	percentage (%)
MEGs	77	52	0.68
PEGs	52	45	0.87
all genes	26521	19333	0.73

**Author response table 2. resptable2:** 

*Arabidopsis thaliana*
	total number of genes	number of genes with PHE1 DNA-binding motifs	percentage (%)
MEGs	145	97	0.69
PEGs	150	123	0.82
all genes	33323	23784	0.71

**Author response table 3. resptable3:** 

p-values	*Capsella rubella*	*Arabidopsis thaliana*
PEGs vs. all genes	0.0269	0.0040
MEGs vs. all genes	0.2896	0.2329